# Assessing empathy in adults: A Malay language validation and measurement invariance of the Empathy Quotient (M-EQ)

Su Woan Wo[1], Ria Heryani Mumfahir[1], Cher Yi Tan[2], Louise Phillips[3], Min Hooi Yong[4]*

1 School of Psychology, Faculty of Medical and Life Sciences, Sunway University, Sunway City, Malaysia, 2 Department of Psychology, HELP University, Kuala Lumpur, Malaysia, 3 School of Psychology, University of Aberdeen, Old Aberdeen, United Kingdom, 4 Department of Psychology, University of Bradford, Bradford, United Kingdom

* m.yong@bradford.ac.uk

## Abstract

The Empathy Quotient (EQ) is one of the most frequently used scales to measure empathy in adults and has been translated into many languages. However, there is still much debate on its dimensionality from its initial inception. Further, we know very little regarding EQ's measurement invariance to ensure that the instrument EQ measures the same construct consistently across different groups (e.g., gender, language type) and having meaningful comparisons without bias. In this study, we validated a Malay version of the Empathy Quotient (M-EQ), a self-report questionnaire among Malaysian adults and examined its measurement invariance across gender and language type. We recruited 586 participants aged between 18–53 years ($M_{age}$ = 23.98, $SD$ = 5.89) to complete the M-EQ. Our confirmatory factor analysis results showed that the M-EQ had two factors (emotional empathy, cognitive empathy) with 23 items retained. The internal consistency and test-retest reliability of the M-EQ was acceptable. Measurement invariance testing indicated that the two-factor structure was broadly comparable across gender and language groups, with configural and metric invariance supported, although some indices suggested cautious interpretation. Compared to males, females scored higher than males on total M-EQ and both subscales. Taken together, the findings support a psychometrically sound 23-item version of the EQ in Malay-speaking adults with evidence for internal structural validity, which may be useful for research and applied contexts where brief and culturally appropriate measures of empathy are required. Overall, our findings contribute to ongoing debates on the dimensionality of the EQ, with more research needed to establish convergent and criterion-related validity of the M-EQ.

**Data availability statement:** All data underlying the findings including the 23-item M-EQ are available in the Open Science Framework repository at https://osf.io/c872d.

**Funding:** This work was supported by Newton Fund Institutional Links grant ID: 331745333, under Newton-Ungku Omar Fund partnership to LP and MHY. The grant is funded by the UK Department for Business, Energy and Industrial Strategy and Malaysian Industry-Government Group for High Technology (MIGHT) and delivered by the British Council. For further information, please visit www.newtonfund. ac.uk There was no additional external funding received for this study.

**Competing interests:** The authors have declared that no competing interests exist.

## Introduction

Empathy is commonly defined as an intricate set of cognitive and affective processes that generates individuals' capability or tendency to recognize, understand, vicariously experience, and respond to others' emotions [1,2]. Conceptually, cognitive empathy is typically characterized as the mental ability to infer others' mental states (i.e., thoughts and feelings) and to comprehend how one's actions may influence others' emotions. Meanwhile, affective empathy is conceptualized as a trait that promotes inclination to share and vicariously experience others' emotional states [1]. One commonly used instrument is the Empathy Quotient (EQ) scale [3,4]. The authors reported that the EQ could identify gender and group differences in both the neurotypical population and in autistic groups and that the EQ measures both cognitive and affective components simultaneously [3,4].

One systematic review of 50 psychological studies examined 23 instruments to assess empathy [5]. Their results showed that the EQ was the most popular, with 11 studies using them. However, there was variation in the versions used: six used the 60-item version (40 EQ plus 20 filler items), three used the 40-item version, and one study each used the 22-item and 15-item versions. In terms of internal consistency, the authors reported that the highest Cronbach alpha was for the 40-item version (0.84) and the 60-item was the lowest at 0.76. When it comes to the structure in the instrument, it is important to note that most studies did not have a consistent factor model. Their analysis showed a range between one-factor to four-factor model for the EQ, and this is partly attributed to the low goodness of fit indexes in some studies with no consensus on what is the best factorial structure. Other researchers analysed the original EQ [3] and reported that it was a single factor. So far, only one study reported a single factor but this was based on a 15-item EQ in Mandarin [6]. Zhao et al. [6] suspected that cultural definitions may have blurred the distinction between emotional and cognitive empathy in the Chinese sample, noting that Chinese people are not encouraged to verbalise their feelings. The cultural norm of limited emotional expression could also apply broadly in Asia. For example, one study showed that young and older adult Malaysian performed poorly in a Theory-of-Mind task, in which the authors attributed this to a reduced tendency to explicitly express mental states compared to Western participants [7]. However, most studies reported a three-factor structure of the EQ irrespective of number of items and language of translation, and that the 3-factor model is often labelled as "cognitive empathy", "emotional reactivity" and "social skills" [4,8–13]. The first factor "cognitive empathy" reflects the ability to recognise and understand others' affective states, e.g., "I can tell if someone is masking their true emotion – item 36). The second factor "emotional reactivity" measures emotional reactions of other people rather than their own, e.g., "seeing people cry doesn't really upset me – item 19". Finally, the third factor "social skills" refers to applying social conventions, e.g., "I often find it difficult to judge whether something is rude or polite – item 8". Taken together, the 3-factor model indicates that the EQ-40 requires individuals to not only understand the thought processes (cognitive) and emotions (affective) but also their social skills and judgment to be applied in social situations (social).

The large number of EQ versions translated to multiple languages attest to the fact that EQ is simple to use and encompass the multifaceted dimension of empathy. Table 1 expands the [5]'s systematic review and has the translated versions with information pertaining to its consistency, reliability, number of items in the translated instrument, identified factor models, demographics, and gender difference. As our interest was to perform validation on the Malay version of the EQ, we reviewed empirical studies that included a translation to local language from the original English version. For this

**Table 1. Overview of the psychometric properties of the Empathy Quotient (EQ) across various countries.**

| Study | Country/ Language | Internal Consistency (Cronbach alpha) | Test-retest Reliability (Pearson's r) | Sample Size / Mean age in years of whole sample (SD) | EQ scores | | Factor model | Number of items | Test-retest Interval |
|---|---|---|---|---|---|---|---|---|---|
| | | | | | Female M (SD) | Male M (SD) | | | |
| [3] – Control group | UK/English | .92 | .97 | 197 / 34.20 (11.80) | 47.2 (10.2) | 41.8 (11.2) | n.r | 60 | 12 months |
| [4] | UK/English | n.r | .84 | 172 / 34.10 (10.40) | 49.6 (9.6) | 40.9 (11.9) | 3 | 28 | 10-12 months |
| [8] | UK/English | .85 | n.r | 348 / 26.90 (12.27) | 46.3 (9.5) | 37.9 (10.5) | 3 | 15 | n.r |
| [14] – Control Group | Japan / Japanese | .86 | n.r | 137 / 29.60 (4.46) | 36.9 (10.73) | 31.1 (10.7) | n.r | n.r | n.r |
| [14] – Student Group | Japan / Japanese | .86 | n.r | 1,250 / 19.40 (1.35) | 36.1 (10.44) | 30.6 (9.92) | n.r | n.r | n.r |
| [16] | France / French | .81 | .93 | 410 / 21.00 (3.24) | 41.4 (7.7) | 37.7 (10.0) | 3 | 60 | 6-24 weeks |
| [17] | Canada / French | .83 | n.r | 100 Grp1: 22.86 (4.42) Grp2: 21.96 (2.65) | Group 1: 43.5 (5.5) Group 2: 41.9 (10.1) | Group 1: 38.4 (9.2) Group 2: 34.2 (6.8) | n.r | n.r | n.r |
| [9] | Korea / Korean | .78 | .84 | 478 / 27.20 (n.r) | 35.8 (9.2) | 34.7 (10.5) | 3 | 28 | 4 weeks |
| [10] | Italy / Italian | .79 | .85 | 256 / 24.00 (4.50) | 45.5 (9.3) | 41.8 (9.4) | 3 | 60 | 4 weeks |
| [18] | Italy / Italian | .68 to .74 | n.r | 633 / 24.30 (5.90) | 17.6 (5.2) | 15.8 (4.4) | 3 | 15 | n.r |
| [13] | Russia / Russian | .85 | .94 | 221 24.90 (7.70) | 43.5 (11.2) | 40.2 (10.7) | 3 | 21 | 2 weeks |
| [11] | Netherlands / Dutch | .89 | .78 | 685 / 33.00 (14.50) | 49.0 (10.4) | 39.1 (12.0) | 3 | 28 | 6-20 months |
| [19] | China / Mandarin | .86 | .82 | 588 / 24.12 (6.20) | 39.6 (10.3) | 37.1 (10.4) | 1 | 15 | 3-4 weeks |
| [12] | Turkey / Turkish | .76 | .95 | 436 / 22.60 (7.23) | 46.4 (8.7) | 43.7 (8.8) | 3 | 29 | 2 weeks |
| [15] | Brazil / Portuguese | .58 − .68 | n.r | 237 / 31.00 (14.09) | n.r | n.r | 3 | 15 | n.r |
| [20] | China/ Uyghur | .82 | .90 | 1638 / 21.73 (3.24) | 25.7 (0.5) | 22.2 (0.4) | 4 | 29 | 1 week |
| Current Study † | Malaysia | .88* | .81 | 354 / 26.19 (6.65) | 26.79 (8.18) | 22.40 (7.16) | 2 | 23 | 4 weeks |

N = sample size, M = mean, SD = standard deviation, n.r = not reported. Grp = group. † Sample 2 only. * Omega reliability.

study, our inclusion criteria for Table 1 were (1) neurotypical population, (2) adults aged 18 and older with the oldest as 84 years old reported in [11], and (3) studies were published in English. We identified published studies via PubMed and Google Scholar using empathy, empathy quotient, adults in our search terms. Most studies recruited university students, but some recruited from general population, e.g., [13–15] and participant databases [3,11].

## Research aim

At present, there is no Malay version of the EQ. The Malay language is the national language in Malaysia, Indonesia, Brunei, and Singapore with approximately 220 million speakers. It is also widely spoken in southern Thailand, Philippines, and East Timor [21]. For this study, our aim was to develop and validate a Malay language version of the EQ (M-EQ). Based on the inconsistent factor models reported in previous reviews (see Table 1 and [5]), our first objective was to examine whether the M-EQ aligns better with a 1-factor as per initial inception [3] or any other structure model reported in Table 1. Our second objective was to assess its temporal reliability, and thus we conducted a test-retest over a 4-week interval. Additionally, our third objective was to evaluate measurement invariance (MI) on two variables; language types and gender.

Regarding language type, this study was conducted in a multicultural and multilingual context with both Malay and English taught formally in Malaysian schools. However, other languages such as Chinese Mandarin (a Sino-Tibetan language) and Tamil (a Dravidian language) are also widely used and form part of the vernacular education system. Unlike Malay and English, which use the Latin alphabet, Chinese and Tamil employ logograms and a syllabic script (abugida) respectively. These differences in writing and cultural backgrounds may influence how individuals perceive words, comprehend sentences and subsequently think about it [22,23].

In terms of gender, past studies have shown that female participants typically had higher empathy scores compared to males, across the lifespan and cultures (see Table 1 for details). Many researchers postulated that higher levels of empathy in females, coupled with higher nurturing traits, is particularly useful for social conflict resolution and serves as a protective factor as they progress through childhood [24]. Baez et al [25]'s study reported that females scored higher on self-report assessment but had no meaningful difference in an experimental paradigm. They argued that self-report measures is a reflection in how empathic women and men like to appear, thus favouring gender stereotypes in that women are highly empathic. Taken together, it remains a question whether there is a female advantage on empathic response, but likely to appear as female superiority on self-report measures on affective items. Therefore, our fourth objective was to examine whether M-EQ measures empathy equally across both genders.

## Materials and methods

### Participants

We recruited a total of 586 participants; 235 males and 351 females with a mean age of 23.98, $SD = 5.89$. Sample 1 had 232 participants, while Sample 2 had 354 participants. See Table 2 for detailed demographic information. Our participants were recruited through university notices and word of mouth references. Participants with confirmed neurodegenerative, psychiatric disorders, and vision and hearing impairments were excluded from the study. We also screened our participants for potential autism and found none to have scored above 6 in the Autism-Quotient (AQ-10) [26]. All participants completed a set of questionnaires: demographic questionnaire and M-EQ. Sample 1 participants completed the survey voluntarily while Sample 2 participants were given a cinema ticket worth ~USD3 for participating. Our first data collection was completed in February 2019 while the second data collection was completed in July 2022. We stopped data collection for two years due to the pandemic movement restrictions (2020–2021). We obtained institutional ethical approval (SUREC2018/040 and PGSUREC 2019/006) and all participants provided written consent before commencing the study.

**Table 2. Demographic characteristics of total sample (n = 586).**

| | Total N = 586 (%) | Sample 1 (*n* = 232) | Sample 2 (*n* = 354) |
|---|---|---|---|
| Gender | | | |
| Male | 235 (40.1) | 117 (50.4) | 118 (33.3) |
| Female | 351 (59.0) | 115 (49.6) | 236 (66.7) |
| Age range (years) | | 18–25 | 18–53 |
| Mean age (years ± *SD*) | 23.98 ± 5.89 | 20.60 ± 1.17 | 26.19 ± 6.65 |
| Ethnicity | | | |
| Malay | 264 (45.1) | 76 (32.8) | 188 (53.1) |
| Chinese | 254 (43.3) | 123 (53.0) | 131 (37.0) |
| Indian | 40 (6.8) | 14 (6.0) | 26 (7.3) |
| Others | 28 (4.8) | 19 (8.2) | 9 (2.5) |
| Highest level of education | | | |
| Completed secondary school | | | 69 (19.5) |
| Completed undergraduate degree | | | 231 (65.3) |
| Completed postgraduate degree | | | 52 (14.7) |
| Did not complete secondary school | | | 2 (0.6) |
| Language proficiency | | | |
| Malay and/or English | 356 (60.8) | 122 (52.6) | 234 (66.1) |
| Chinese and/or Tamil dialects | 230 (39.2) | 110 (47.4) | 120 (33.9) |

Numbers in brackets are percentages; *SD* = standard deviation.

## Translation to Malay and back-translation to English

Translation of the EQ English version to Malay was performed by two research assistants according to international guidelines [27], and later back-translated to English with another two research assistants. The final Malay version was selected by the first three authors, as all were fluent and/or native in both languages. They read and evaluated whether the translated items were comprehensible and made recommendations for modifications.

## Empathy quotient (EQ)

This measure had 40 items [3] and participants selected one of four response choices; 'strongly agree', 'slightly agree', 'slightly disagree', 'strongly disagree'. See S1 File for the M-EQ. An example of the item is item 14 "I am good at predicting how someone will feel", the Malay version was "Saya berkebolehan dalam meneka perasaan orang lain". In order to avoid response bias, half of the items were phrased to yield 'agree' answers while the other half of the items were phrased to yield 'disagree' answers for empathic response. The most appropriate answer choices that indicate empathic response (i.e., 'agree' or 'disagree' that indicate empathic response) scored 2 points, whereas the less appropriate answer choices scored 1 point. Inappropriate answer choices for empathic response will not generate any point for participants. A higher score indicated higher empathy.

## Procedure

All participants completed both the demographic and M-EQ online in university psychology labs. Sample 2 participants returned to the lab to complete the task a second time after four weeks. They were informed to answer as honestly as they can and to take as much time as needed. Note that Sample 2 participants also participated in other studies not reported here.

## Analytical plan

We checked for normality in the M-EQ using Shapiro-Wilk and visual inspection of the histogram. All data were normally distributed (Shapiro-Wilk $ps > .05$). Factor analysis was conducted to test, explore, and confirm the factorial structure and measurement invariance of the M-EQ. The reliability of the model will be determined once the underlying factors are extracted. A Kaiser–Meyer–Olkin (KMO) value above .70 and a significant Bartlett's test ($p < .05$) indicated the data is suitable for factor extraction.

We implemented a 3-step validation process. In Step 1, we tested a one-factor model using confirmatory factor analysis (CFA) with AMOS 27 in Sample 1 participants. We then conducted an exploratory factor analysis (EFA) in R to better understand the underlying structure (Step 2). The number of factors to retain was determined using parallel analysis, which compares observed eigenvalues with those obtained from random data [28]. Maximum likelihood extraction with oblique rotation was applied, given the expected correlations among factors. Based on the EFA results, a two-factor model was proposed to which we tested this using CFA on Sample 2 participants to confirm its structural validity (Step 3). This sequential approach was adopted to minimize data-driven model modification and to ensure that confirmatory analyses were grounded in an empirically derived factor structure.

To determine acceptable standards, we used the following guidelines for root mean square error of approximation (RMSEA) of 0.06 or lower (with 90% confidence interval below 0.08), and Standardized Root Mean Square Residual (SRMR) of 0.09 or lower [29], chi-square/df ratio value less than 3, comparative fit index (CFI) and Tucker-Lewis Index (TLI) greater than 0.9.

Internal consistency of the M-EQ was estimated using Cronbach's α and McDonald omega with a value >0.7 indicates satisfactory, and test-retest reliability using Pearson's correlation coefficients. Note that Sample 2 participants completed the M-EQ again after a four-week interval. Gender differences on the empathy scores were analysed using independent-samples *t*-tests.

Once the construct validity of the M-EQ is determined, we evaluated measurement invariance following a 2-stage analytic strategy grounded primarily in CFA. We conducted multi-group CFA to examine invariance across gender and language types using WLSMV estimator in R (*lavaan* package). Measurement invariance was assessed sequentially. First, we tested configural invariance to determine whether the two-factor structure was comparable across groups. Next, we tested metric invariance by constraining factor loadings to equality. Model fit was evaluated using CFI, RMSEA, and SRMR, and changes in fit were judged using recommended criteria (ΔCFI ≤ .010, ΔRMSEA ≤ .015, ΔSRMR ≤ 0.030) [30]. Given known sensitivity of SRMR in WLSMV models, this index was interpreted cautiously.

To supplement the CFA results and to explore potential item-level sources of misfit, we conducted item response theory–based differential item functioning (DIF) analyses. These analyses were intended as diagnostic tools to identify whether specific items functioned differently across groups. Separate multigroup graded response models were estimated for the EE and CE subscales to align with the established two-factor structure. Anchor items were selected a priori and used to link groups to a common latent metric. DIF was evaluated using likelihood-ratio tests comparing nested models, with Benjamini–Hochberg correction applied to control for false discovery. Expected a posteriori (EAP) reliability was computed to assess the precision of latent trait estimates.

## Results

### Factor analysis

Based on the original EQ-40 [3], we examined whether the M-EQ data fit a 1-factor-model using Sample 1 participants ($n = 232$). Results showed an inadequate model of fit for 1-factor: $\chi^2$ (740) = 1901.8, $p < .001$, $\chi^2/df = 2.57$, RMSEA = 0.082, SRMR = 0.1177; CFI = .449; TLI = .419. These results demonstrated a need for considerable improvement in the fit between the model and the data to establish the concept of empathy among the Malaysian sample. Therefore, an EFA was conducted to explore the underlying structure of the EQ-40 from the Malaysian sample.

We conducted EFA in sample 1 ($n = 232$) in R. EFA was performed using maximum likelihood with oblimin rotation, given the expected correlations among factors. The data were suitable for factor analysis with KMO = 0.812, Bartlett $p < .001$. Factor retention was guided using Parallel Analysis. We used the following criteria: items were retained if they showed factor loading ≥ .40. Items were considered for removal when the items show cross-loading ≥ .30 on multiple factors and low communality (h²) < .20–.30 [31,32].

Parallel analysis initially supported a four-factor solution with an acceptable overall model quality (RMSEA = 0.037, RMSR = 0.05, TLI = 0.87). However, closer inspection of item-level diagnostics indicated that this solution lacked psychometric and conceptual coherence. Several items have low communalities and some items have high cross-loading across factors indicating poor conceptual and psychometric coherence. Specifically, Factor 3 was defined by a single item (EQ27) with strong loading (λ = 0.82, h² = 0.68) while other items (EQ29, EQ40, and EQ39) showed weak loadings (0.32–0.36) and cross-loaded on Factor 4. A latent factor defined by only one item is considered psychometrically inadequate and was therefore deemed unacceptable. Similarly, items loading on factor 4 (EQ21, EQ22, EQ39 and EQ 25) showed weak factor loading (0.39 to 0.37 with cross loading), thus undermining this factor. Taken together, these findings suggest that the 4-factor solution represented an over extraction of factors and thus rejected.

We then removed seven items (EQ2, EQ3, EQ4, EQ17, EQ23, EQ24, and EQ25) for consistently low loadings across factors. EFA was re-estimated and parallel analysis was conducted again. Findings indicated a 3-factor solution with improved overall model quality (RMSEA = 0.041, RMSR = 0.05, TLI = 0.88). However, factor 3 remained psychometrically invalid with only EQ27 (factor loading of 1.00), indicating statistical over-extraction.

Given these findings, we evaluated a 2-factor solution and removed item EQ27 which was supported by parallel analysis. Model fit indices remained acceptable (RMSEA = 0.042, RMSR = 0.06, TLI = 0.87), and both factors were statistically acceptable. Factor 1 comprised of 13 items (EQ1, EQ11, EQ13, EQ14, EQ15, EQ22, EQ26, EQ28, EQ29, EQ34, EQ35, EQ36, and EQ38) with factor loadings ranging from 0.409–0.701. These 13 items were about the sharing an emotional state, a willingness to help others, and feeling distress or upset to another's pain. An example was "I can pick up quickly if someone says one thing but means another – Item 11". In sum, this factor was conceptualised as "emotional empathy" (EE). Factor 2 also comprised of 13 items (EQ5, EQ7, EQ8, EQ10, EQ12, EQ16, EQ18, EQ19, EQ20, EQ30, EQ31, EQ32, and EQ33) with factor loadings ranging from 0.411 to 0.580. These items contain perspective-taking, understand other's people feelings, and imagined what was others' perspectives. Example of an item is "I often find it difficult to judge if something is rude or polite – item 8". Hence, this factor was conceptualised as "cognitive empathy" (CE). See Table 3 for details.

To confirm the structural validity of M-EQ with 26 items, we repeated CFA for 2-factor model with Sample 2 participants ($n = 354$) using weighted least squares mean- and variance-adjusted (WLSMV) estimation. The model specification was informed by the EFA results, and 14 items previously identified as problematic were excluded prior to model estimation. Our results showed an adequate model of fit: $\chi^2(298) = 671.96$, $p < .001$; $\chi^2/df = 2.25$, CFI = 0.963, TLI = 0.960, RMSEA = 0.060 (90% CI .054–.066), SRMR = 0.074. Factor 1 (EE) was very strong, with all items loading above .50. Factor 2 (CE) showed acceptable fit, but three items (EQ10, 32, 33) exhibited very weak standardized loadings (< .30), indicating they did not reliably measure the latent construct in this sample. As such, these three items were removed and the CFA was re-estimated.

The refined two-factor model (23 items) demonstrated good fit to the data, $\chi^2(229) = 528.40$, $p < .001$; $\chi^2/df = 2.31$; CFI = .970; TLI = .967; RMSEA = .061 (90% CI [.054, .068]); SRMR = .072. Standardized factor loadings ranged from 0.51 to 0.77 for Factor 1 (EE) and from 0.35 to 0.76 for Factor 2 (CE). No post-hoc model modifications were applied. Although EQ19 showed a relatively lower standardized loading (0.353), it exceeded the minimum acceptable threshold and was retained to preserve content coverage of Factor 2. In addition, removing EQ19 did not result in a meaningful improvement in overall model fit.

We next estimated a second-order CFA model in which EE and CE factors were loaded onto a higher-order empathy construct using R (Lavaan package). The model did not converge to a proper solution, and produced a non-invertible

**Table 3. Factor loadings for M-EQ from EFA using the oblimin method (n = 232).**

| Item | Factor loadings | |
|---|---|---|
| | **F1** (Emotional Empathy Subscale) | **F2** (Cognitive Empathy Subscale) |
| EQ14 | 0.71 | |
| EQ34 | 0.71 | |
| EQ36 | 0.64 | |
| EQ15 | 0.60 | |
| EQ22 | 0.59 | |
| EQ26 | 0.53 | |
| EQ35 | 0.54 | |
| EQ38 | 0.54 | |
| EQ1 | 0.56 | |
| EQ28 | 0.48 | |
| EQ13 | 0.45 | |
| EQ11 | 0.43 | |
| EQ29 | 0.41 | |
| EQ18 | | 0.65 |
| EQ20 | | 0.60 |
| EQ30 | | 0.59 |
| EQ31 | | 0.59 |
| EQ33 | | 0.55 |
| EQ19 | | 0.54 |
| EQ16 | | 0.56 |
| EQ10 | | 0.51 |
| EQ12 | | 0.49 |
| EQ32 | | 0.48 |
| EQ8 | | 0.47 |
| EQ7 | | 0.43 |
| EQ5 | | 0.41 |

information matrix and a negative variance estimate for the second-order factor, indicating that the data does not support a unidimensional structure. Therefore, the correlated 2-factor model was retained for as the appropriate structural representation of the M-EQ, $\chi^2(229) = 481.206$, CFI = 0.973, TLI = 0.971, RMSEA = 0.056, SRMR = 0.070, with the latent factors showing a moderate negative correlation (r = −0.434, $p < .001$).

## Measurement invariance (MI) for M-EQ

**Gender.** Measurement invariance across gender was first examined using single-group CFAs. Single-group CFA indicated acceptable fit in the female group (CFI = 0.975, TLI = 0.972, RMSEA = 0.061, SRMR = 0.080), and the male group also showed acceptable CFI, TLI, and RMSEA values but elevated SRMR (CFI = 0.944, TLI = 0.938, RMSEA = 0.067, SRMR = 0.103). We next proceed with the configural model, and it demonstrated acceptable overall fit across gender (CFI = 0.968, TLI = 0.965, RMSEA = 0.063, SRMR = 0.088), indicating that the two-factor structure was broadly similar across male and female participants. Constraining factor loadings to equality across gender resulted in acceptable model fit (CFI = 0.959, RMSEA = 0.070, SRMR = 0.093), with minimal changes in fit indices (ΔCFI = −0.009, ΔRMSEA = +0.007), supporting metric invariance. However, the slightly elevated SRMR values warrant cautious interpretation.

We further examined gender-related item level equivalence using Item Response Theory-based Differential Item Functioning (DIF) analyses. First, we examined a posteriori (EAP) reliability for both subscales and it was high for EE (0.865) and acceptable for CE (0.775), supporting the use of latent scores for group comparisons. DIF analyses revealed no evidence of gender-related DIF for either EE or CE. Likelihood-ratio DIF tests were conducted for discrimination (a1) and threshold (d1–d3) parameters, with $p$-values adjusted using the Benjamini–Hochberg false discovery rate procedure. No items showed statistically significant DIF and none of the drop-in-fit tests indicated meaningful improvements in model fit. These indicate invariant item functioning across gender groups (see Table 4).

**Language types.** We repeated the analyses using the sample 2 (Latin: English/Malay = 234, non-Latin: Chinese/Tamil = 120). Single-group CFA indicated acceptable fit for the Latin group (CFI = 0.964, RMSEA = 0.062, SRMR = 0.082), whereas the non-Latin group showed acceptable but slightly poorer fit for CFI particularly in terms of RMSEA and SRMR (CFI = 0.957, RMSEA = 0.089, SRMR = 0.108). The configural model demonstrated acceptable overall fit (CFI = 0.961, RMSEA = 0.072, SRMR = 0.091), indicating that the two-factor structure was broadly similar across language groups, although model fit was not optimal. Constraining factor loadings to equality resulted in minimal changes in model fit (metric model: CFI = 0.953, RMSEA = 0.077, SRMR = 0.096; ΔCFI = −0.008, ΔRMSEA = +0.005), suggesting support for metric invariance across language groups. However, given the elevated SRMR values and poorer fit in the non-Latin group, these findings should be interpreted with caution.

DIF analyses were conducted to further examine potential item-level non-invariance. EAP reliability estimates again indicated adequate measurement precision for latent mean comparisons (EE = 0.862; CE = 0.767). DIF analyses showed no evidence of differential item functioning for either EE or CE across language groups. Likelihood-ratio DIF tests for discrimination (a1) and threshold (d1–d3) parameters, with Benjamini–Hochberg FDR correction, yielded no significant item-level differences (all Δ−2LL = 0). These results indicate invariant item functioning across language types, suggesting that M -EQ items function equivalently for respondents using Latin and non-Latin as their first language (see Table 4).

## Internal consistency and test-retest coefficients

Internal consistency of the M-EQ 23 items was evaluated using both Cronbach's α and McDonald's $\omega_t$. The EE subscale demonstrated good reliability (α = .86, $\omega_t$ = .86), and the CE subscale showed acceptable reliability (α = .78, $\omega_t$ = .78). As the scale exhibited a multidimensional structure, Cronbach's alpha for the total score was lower (α = .67), which is expected for scales comprising multiple correlated dimensions, as alpha assumes unidimensionality and equivalent factor loadings. When these assumptions are violated, alpha tends to underestimate reliability [33,34]. Therefore, McDonald's omega was also computed for the total score, yielding high composite reliability ($\omega_t$ = .88). Subscale scores are emphasized for interpretation, with the total score reported as a supplementary index.

**Table 4. Item Response Theory- based Measurement Invariance of the EQ Malay Across Gender and Language.**

| Subscales | Emotional Empathy | | | Cognitive Empathy | | |
|---|---|---|---|---|---|---|
| Variable group | No. of items | Anchor items | DIF (Thresholds d1–d3) | No. of items | Anchor items | DIF (Thresholds d1–d3) |
| 1. Gender (Male vs Female) | 13 | EQ14, EQ34, EQ36, EQ22 | Non-detected | 10 | EQ31, EQ8, EQ18, EQ12 | Non-detected |
| 2. Language (Latin vs non-Latin) | 13 | EQ14, EQ34, EQ36, EQ22 | Non-detected | 10 | EQ31, EQ8, EQ18, EQ12 | Non-detected |

*Note*: DIF = differential item functioning. DIF was evaluated using multi-group IRT models with anchor-item linking. Anchor items are invariant reference items used to place groups on a common latent metric, enabling valid detection of DIF in the remaining items. Anchors were selected a priori based on strong factor loadings and conceptual relevance to the grouping variable. No DIF was detected at either the threshold (d1–d3) or discrimination (a1) levels.

Test–retest reliability was examined using Pearson correlations between Time 1 and Time 2 scores. The EE subscale demonstrated good temporal stability ($r = 0.734$, $p < .001$), as did the CE subscale ($r = 0.748$, $p < .001$). The total empathy score showed moderate test–retest reliability ($r = 0.805$, $p < .001$). Overall, these findings indicate satisfactory temporal stability, particularly for the subscale scores.

## Gender differences in empathy

Past studies have shown a gender difference in empathy scores, with female participants scoring higher than males (see Table 1). Our results showed that female participants had overall higher empathy score compared to male participants, $t(352) = 4.96$, $p < .001$, Cohen's $d = 0.56$. Likewise, female participants scored higher on both EE and CE compared to male participants, $t(352) = 3.69$, $p < .001$, Cohen's $d = 0.42$, and $t(352) = 4.59$, $p < .001$, Cohen's $d = 0.52$ respectively] (see Table 5 for details).

## Discussion

To the best of our knowledge, our analyses demonstrated psychometric and theoretical robustness as evidenced from the EFA and CFA showing two factors (Emotional Empathy 'EE', Cognitive Empathy 'CE') and high reliability scores in the 23-item M-EQ. The M-EQ 2-factor model is well aligned with theoretical understanding on how empathy is defined and measured [1,5]. Measurement invariance analyses indicated that the two-factor structure was broadly comparable across gender and language groups. The configural model showed acceptable fit, and changes in fit indices supported metric invariance, although elevated SRMR values suggest a cautious interpretation is warranted. Supplementary IRT-based DIF analyses did not identify any item-level bias across groups, providing additional support for the M-EQ.

We note the differences between our findings and past studies, e.g., factor model, number of items as shown in Table 1. Past studies that validated the EQ in English [4,8] or other languages reported a 1-factor [19], 4-factor [20] but the majority showed a 3-factor model with EE, CE, and Social Skills (SS) [9–13,15,16,18]. Yet many acknowledged issues pertaining to SS and EE. One study showed multiple loading on the same item across two factors [12] and another suspected that the SS factor is better suited to measure personality type, i.e., assertiveness rather than an empathy dimension [15]. Others reported lower consistency and lower intercorrelations in the SS compared to CE and EE [11] and the authors suspected that items in the SS are reversed-scored which may have obscured the intended meaning. Additionally, some argued that the EQ instrument is positively correlated to social desirability because empathizing require individuals to be compliant with expectations from others [10,12,16]. For example, item 12 "It is hard for me to see why some things upset people so much" implies that social desirability may take place instead of actual empathic responding. In fact, some studies showed a positive relation between social skills and Marlowe-Crowne Social Desirability Scale (MC-SDS) (positive attribution and denial) and thus was removed from their analyses [10,16]. However, we retained item 12 in the M-EQ and is categorised as CE in our analyses. This is because we use cognitive mechanisms to infer from one situation to another

**Table 5. Mean scores and standard deviation on the M-EQ by gender (n = 354).**

| Variable | Total (n = 354) | Gender | | t | p |
|---|---|---|---|---|---|
| | | Male (n = 118) | Female (n = 236) | | |
| 23 items M-EQ | 45.00 (25.33) | 22.40 (7.16) | 26.79 (8.18) | 4.96 | <.001 |
| EE | 14.91 (5.30) | 13.47 (4.90) | 15.64 (5.36) | 3.69 | <.001 |
| CE | 10.41 (4.42) | 8.93 (4.23) | 11.16 (4.34) | 4.59 | <.001 |

Note: *t*-value comparing males to females.

and the cognitive aspect is helping us predict behaviours and emotions from others, which then promotes the capacity to adopt another person's perspective [35].

Note that most studies reported correlations between all three factors (CE, EE, SS) but that they were low enough to preserve discriminant validity with *r* values ranging from .23 to .43 [9–11,18]. Many studies also reported mixed results when they tested Muncer and Ling [8] 15-item CFA in their own samples, indicating potential psychometric errors in their analysis. Recent evidence showed that using principal components analysis with eigenvalue > 1 and applying varimax 'orthogonal' rotation could inflate factor counts despite high loadings and large sample size [36]. Further, orthogonal rotation is inappropriate for correlated psychological constructs because Fabrigar et al. [31] argued that when latent variables are correlated, the oblique rotation are more realistic due to the overlapping nature of these constructs. Taken together, there is much room for error in validating the factorial structure in the EQ. To further confirm our 2-factor model, we tested a second-order CFA in which EE and CE were specified to load onto a higher-order empathy factor. The result showed a non-invertible information matrix and negative latent variance, indicating that a unidimensional hierarchical model is not supported. Further, the difference in subscales length (10 items for CE vs 13 items for EE) and that each subscale represented a distinct but related constructs, we can still use the total score since we had high reliability ($\omega_t$ = .88).

We note that the M-EQ has 23 items and this finding is unusual compared to the original that had 40 items. Past studies reported a better fit with 15 items [8,15,18,19], or 28 items [9,11], but they also reported a 3-factor model except for one [19]. The 15-item has been proposed as the brief version of the EQ scale [8] and this version has been validated in other studies with reasonably good reliability [9–11,18]. In the 15-item, five items identified as CE were categorised as EE in the M-EQ while only three out of the five items under EE appeared as CE (item 16, 19, 33). Only two items out of five items in the SS were retained in the M-EQ and they were identified under CE (Item 7, 8). Some studies applied the same items in their analysis but others identified different items for CE, EE and SS respectively [10,11], indicating interpretation of these items may mean differently due to subtle nuances in the translated language, rather than core empathic capacity per se. Taken together, we are confident that our analyses for the 23-item M-EQ is robust psychometrically and very much aligned with theoretical underpinning empathy dimensions. Importantly, all core domains and subcomponents of the original EQ framework remain represented in the final item set, with each domain retaining multiple indicators reflecting its theoretical breadth. The omitted items predominantly consisted of those with weak factor loadings, substantial cross-loadings across domains, or low communalities, indicating limited contribution to the latent constructs in the Malay language context. Further, the omitted items did not represent unique conceptual facets of empathy that were absent from the retained items; rather, they overlapped substantially in content with retained indicators within each domain. As such, item removal reflects refinement of measurement rather than loss of conceptual content, ensuring preservation of essential empathic elements while improving clarity and cultural suitability.

For gender, evidence has shown that females tend to have higher empathic scores compared to males (see Table 1). Our results are concordant with previous studies in that the female participants scored higher on both EE and CE as well as the overall EQ scores. One meta-analysis across 57 countries reported a female advantage in accuracy and rapidity when decoding emotional nonverbal behaviours across the lifespan [37]. One study reported that females rated themselves more superior in a self-report empathy scale compared to males [38]. However when tested in an experimental paradigm, this gender superiority was only present when the motivation for empathy was raised, suggesting that gender roles and stereotypical beliefs may play a role in enhancing empathy [38]. Although stereotypical beliefs may be present during infancy and early childhood, these differences tend to converge in adulthood, indicating that empathic responses are relatively stable in adulthood [39]. In Rochat [40]'s review, she proposed that both males and females have different approaches towards empathy, in that females use more bottom-up 'affective' while males tend to rely more on top-down processing 'cognitive'. Importantly although the M-EQ findings showed female superiority for both CE and EE in our study, the measurement invariance results indicated that the two-factor M-EQ structure operates comparably across genders. While single-group CFA indicated slightly poorer fit in the male subgroup (e.g., elevated SRMR values), the multigroup

 

models supported configural and metric invariance, suggesting that males and females interpret and respond to the items in a structurally similar manner. However, the slightly elevated SRMR indices point to some localized misfit, and full equivalence should not be assumed without caution. We conducted DIF analyses to further examine potential item-level differences. The absence of significant DIF across both discrimination and threshold parameters suggests that individual items function equivalently across males and females. Taken together, these findings provide converging evidence that the M-EQ demonstrates adequate measurement comparability across gender at both the factor and item levels, supporting its use in gender-based comparisons, while recognising the need for cautious interpretation of minor model misfit.

Whereas in terms of language type, the MI results similarly indicated that the M-EQ functions broadly comparably across Malay speakers from Latin- and non-Latin-based language backgrounds, although some caution is warranted. The two-factor structure was supported at the configural level suggesting that the underlying relationships between items and constructs are generally comparable across groups. Metric invariance was also supported based on minimal changes in fit indices, however the relatively poorer model fit (higher RMSEA and SRMR values) observed in the non-Latin group suggests that some items function differently across language backgrounds. To explore further, DIF analyses were conducted as supplementary item-level diagnostics. The absence of significant DIF across both subscales suggests that no individual items demonstrated systematic bias across language groups indicating that the modest misfit is unlikely to be driven by specific items. We suspect that these may reflect broader differences in response patterns, linguistic nuances, or cultural interpretations of empathy-related behaviours.

An individual's first language could have influenced thought thus an indirect effect on how individuals interpret the intended meaning [22,23]. Indeed, some studies reported that responses to an empathy instrument may be shaped by individual personality traits or cultural norms around emotional expression [11,15]. When comparing studies conducted in Asia, the M-EQ scores are similar to Chinese sample [20] but they had 29 items, while the 15-item also in China [19] and 28-item Korean version [9] had higher empathy scores compared to the Malaysians. These differences in total scores may reflect linguistic or cultural difference or could be attributed to the number of retained items in each version. Shorter or more culturally adapted versions often exclude items with lower discrimination, which can inflate average scores. We are cautious that our finding might be over-explaining the complexity and depth surrounding empathy dimensions. Malaysia is a multi-cultural country with many languages spoken but the Malay language is the official language and taught formally in schools. This meant that our participants who were sampled from university campuses would have some Malay language proficiency thus enabling them to understand the nuances in Malay, despite Malay not being their first language.

We acknowledge that there are several limitations in our study. First, the final two-factor, 23-item structure of the M-EQ emerged through an iterative process that included both theory-driven reasoning and data-driven refinement. Although this approach is common in early-stage instrument validation, it raises the possibility that the model may partly reflect sample-specific characteristics. Although we have a large sample, we acknowledge that our sampling was not sufficiently diverse to represent the human lifespan because our sample is entirely young adults. Further, our university student sampling limits the generalisability to the wider population. Another limitation is that we did not directly measure intelligence. While the EQ does require some verbal intelligence [4], 99.4% of our sample completed a minimum of 12 years of formal education, suggesting that they do have at least average intelligence.

Second, although the study provides initial evidence for construct validity based on EFA, CFA, and supplementary IRT-based DIF analyses, the validity evidence is limited in scope because our evaluation focused primarily on the internal structure. We did not include other behavioural measures such as emotion recognition tasks [41] or other empathy scales such as Toronto Empathy Questionnaire [42] to assess convergent, discriminant or predictive validity. As such, conclusions regarding the construct validity of the M-EQ should be considered preliminary and strengthened in future work using multi-method validation approaches and across different samples. While the measurement invariance results support broad comparability of the M-EQ across gender and language groups, the evidence is not unequivocal. While configural and metric invariance were supported based on recommended fit index changes, some single-group and multigroup

models showed elevated SRMR and RMSEA values, particularly in the non-Latin language group. These patterns indicate localized model misfit, and full equivalence should not be assumed without caution.

Lastly, we acknowledge that our data collection was impacted by the COVID-19 pandemic. Some researchers reported that although cognitive and emotional empathy remained intact during the pandemic, there were significant reduction in empathic social skills [43,44], indicating that empathic response could possibly be atypical during the pandemic outbreak and may have continued long after as a new 'norm'.

Although our study focused on neurotypical adult population, we wish to highlight the potential application of the M-EQ in clinical settings. Some proposed that the multifaceted concept of empathy could contribute to a more complete understanding of various clinical populations such as autism [45]. In their study, they used EQ alongside Autism Spectrum Quotient (AQ) and the Relatives Questionnaire (RQ) (retrospective, parent/informant-report version of the Childhood Autism Spectrum Test) to facilitate quicker identification of autistic individuals and reduce diagnostic wait times. Another study reported that people with low emotional empathy was associated with Dark Triad traits, while low cognitive empathy are specifically linked to narcissism and Machiavellianism [46], indicating EQ's potential utility for psychopathy assessments. Scoring low on cognitive empathy factor is also linked to poor social function in schizophrenia patients [47]. Note that in both studies [46,47], the authors focused on specific factors (cognitive or emotional empathy) in the EQ, suggesting that clinicians could streamline assessments by selecting only the most relevant items for their target population without compromising diagnostic validity. These studies underscore the potential for the M-EQ to be adapted for the clinical population in Malaysia, offering a culturally relevant and time-efficient tool for psychological assessment.

## Conclusion

In sum, the final model M-EQ had 23 items with two factors; CE, EE. Future studies should include ASD participants to evaluate the discriminant and criterion validity of the M-EQ, ensuring its ability to distinguish between neurotypical and autistic samples and accurately classify individuals into the appropriate group. Researchers could also test the M-EQ with other behavioural tasks and empathy instruments to determine its convergent, discriminant or predictive validity on capturing empathic responses. Our findings support that M-EQ is psychometrically sound to measure empathy and is specifically useful in the Malay-speaking community. Our findings further support the cross-cultural stability of the EQ.

## Supporting information

**S1 File. 23-item EQ.**
(DOCX)

## Acknowledgments

This paper is part of RM MSc in Psychology by Research. RM MSc was supervised by MHY and LP.

## Author contributions

**Conceptualization:** Su Woan Wo, Ria Heryani Mumfahir, Louise Phillips, Min Hooi Yong.

**Data curation:** Su Woan Wo, Ria Heryani Mumfahir, Cher Yi Tan.

**Formal analysis:** Su Woan Wo, Ria Heryani Mumfahir, Cher Yi Tan, Min Hooi Yong.

**Funding acquisition:** Louise Phillips, Min Hooi Yong.

**Supervision:** Louise Phillips, Min Hooi Yong.

**Writing – original draft:** Ria Heryani Mumfahir.

**Writing – review & editing:** Su Woan Wo, Louise Phillips, Min Hooi Yong.

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
