## [Decision Letter · Decision Letter 0]

23 Sep 2025

PONE-D-25-12858Assessing empathy in adults: A Malay language validation and measurement invariance of the Empathy Quotient (M-EQ)PLOS ONE

Dear Dr. Yong,

Thank you for submitting your manuscript to PLOS ONE. After careful consideration, we feel that it has merit but does not fully meet PLOS ONE’s publication criteria as it currently stands. Therefore, we invite you to submit a revised version of the manuscript that addresses the points raised during the review process.

**The paper is well-written and convey a clear rationale of study, method, result, and discussion. Please carefully see comments from two reviewers especially in statistical parts. We are looking forward to seeing the revision of this paper.** 

We look forward to receiving your revised manuscript.

Kind regards,

Juthatip Wiwattanapantuwong

Guest Editor

PLOS ONE

**Journal Requirements:**

1. When submitting your revision, we need you to address these additional requirements. Please ensure that your manuscript meets PLOS ONE's style requirements, including those for file naming. The PLOS ONE style templates can be found at https://journals.plos.org/plosone/s/file?id=wjVg/PLOSOne_formatting_sample_main_body.pdf and https://journals.plos.org/plosone/s/file?id=ba62/PLOSOne_formatting_sample_title_authors_affiliations.pdf 2. Please include a complete copy of PLOS’ questionnaire on inclusivity in global research in your revised manuscript. Our policy for research in this area aims to improve transparency in the reporting of research performed outside of researchers’ own country or community. The policy applies to researchers who have travelled to a different country to conduct research, research with Indigenous populations or their lands, and research on cultural artefacts. The questionnaire can also be requested at the journal’s discretion for any other submissions, even if these conditions are not met.  Please find more information on the policy and a link to download a blank copy of the questionnaire here: https://journals.plos.org/plosone/s/best-practices-in-research-reporting. Please upload a completed version of your questionnaire as Supporting Information when you resubmit your manuscript. 3. Thank you for stating in your Funding Statement: This work was supported by Newton Fund Institutional Links grant ID: 331745333, under Newton-Ungku Omar Fund partnership to LP and MHY. The grant is funded by the UK Department for Business, Energy and Industrial Strategy and Malaysian Industry-Government Group for High Technology (MIGHT) and delivered by the British Council. For further information, please visit https://www.newtonfund.ac.uk   Please provide an amended statement that declares *all* the funding or sources of support (whether external or internal to your organization) received during this study, as detailed online in our guide for authors at http://journals.plos.org/plosone/s/submit-now. Please also include the statement “There was no additional external funding received for this study.” in your updated Funding Statement. Please include your amended Funding Statement within your cover letter. We will change the online submission form on your behalf. 4. Thank you for stating the following in the Acknowledgments Section of your manuscript: This work was supported by Newton Fund Institutional Links grant ID: 331745333, under Newton-Ungku Omar Fund partnership to LP and MHY. The grant is funded by the UK Department for Business, Energy and Industrial Strategy and Malaysian Industry-Government Group for High Technology (MIGHT) and delivered by the British Council. For further information, please visit https://www.newtonfund.ac.uk This paper is part of RM MSc in Psychology by Research and her MSc was funded by Sunway University Postgraduate by Research Scholarship. RM MSc was supervised by MHY and LP. We note that you have provided funding information that is not currently declared in your Funding Statement. However, funding information should not appear in the Acknowledgments section or other areas of your manuscript. We will only publish funding information present in the Funding Statement section of the online submission form. Please remove any funding-related text from the manuscript and let us know how you would like to update your Funding Statement. Currently, your Funding Statement reads as follows: This work was supported by Newton Fund Institutional Links grant ID: 331745333, under Newton-Ungku Omar Fund partnership to LP and MHY. The grant is funded by the UK Department for Business, Energy and Industrial Strategy and Malaysian Industry-Government Group for High Technology (MIGHT) and delivered by the British Council. For further information, please visit https://www.newtonfund.ac.uk   Please include your amended statements within your cover letter; we will change the online submission form on your behalf. 5. Please note that your Data Availability Statement is currently missing the direct link to access each database. If your manuscript is accepted for publication, you will be asked to provide these details on a very short timeline. We therefore suggest that you provide this information now, though we will not hold up the peer review process if you are unable. 6. If the reviewer comments include a recommendation to cite specific previously published works, please review and evaluate these publications to determine whether they are relevant and should be cited. There is no requirement to cite these works unless the editor has indicated otherwise.

Reviewers' comments:

Reviewer's Responses to Questions

**Comments to the Author**

1. Is the manuscript technically sound, and do the data support the conclusions?

Reviewer #1: Yes

Reviewer #2: Partly

2. Has the statistical analysis been performed appropriately and rigorously? 

Reviewer #1: Yes

Reviewer #2: No

3. Have the authors made all data underlying the findings in their manuscript fully available?

Reviewer #1: Yes

Reviewer #2: No

4. Is the manuscript presented in an intelligible fashion and written in standard English?

Reviewer #1: Yes

Reviewer #2: Yes

5. Review Comments to the Author

**Reviewer #1:** Sunday, June 1, 2025

Dear authors

Thank you for your good manuscript, entitled "Assessing empathy in adults: A Malay language validation and measurement invariance of the Empathy Quotient (M-EQ)".

This is a valuable study. Here are my comments.

1. Title should change to “Empathy: Measurement invariance of the Empathy Quotient among Malaysian adults”.

2. Keywords should change to “Empathy, Empathy Quotient (EQ), Measurement invariance, Psychometrics; Malaysia”.

3. Authors should add research questions/assumptions of their study in the end of Introduction section.

4. Authors should write all items for the EQ in the Table (All statements).

5. Authors should write clinical implications of their study.

6. Authors should write doi for all references.

Best

**Reviewer #2:** I appreciate the authors’ effort to validate the Malay version of the Empathy Quotient (M-EQ) and to examine its measurement invariance across gender and language type. However, the manuscript is not yet easy to follow, and several conceptual, methodological, and reporting issues currently limit its scientific contribution and clarity.

The manuscript lacks conceptual coherence. The introduction emphasizes cognitive and affective empathy, but the final model includes three factors (cognitive, emotional, and social skills) without adequate justification for the additional factor. Terminology is also inconsistent (e.g., “emotional” vs. “affective” empathy; “sex” vs. “gender”), which reduces clarity. The rationale for including “social skills” as a separate factor and its theoretical grounding should be explained more thoroughly.

It is unclear how Table 1 was compiled. Did the authors conduct their own review, or was the table adapted from previous work? The methodology for obtaining the data in Table 1 should be clarified. In addition, the table would be more useful if it included information about the populations studied (e.g., clinical vs. general population), age ranges, and other relevant demographics.

The manuscript does not adequately justify the use of two samples. It is unclear why the authors treated the datasets as separate for some analyses and combined them for others (e.g., measurement invariance). In addition, because data collection was interrupted by COVID-19, the comparability of pre- and post-pandemic data should be examined. The authors should report whether participants recruited before versus after the pandemic differed on key demographic or psychometric variables, as such differences could affect the factor structure and reliability estimates.

The analytical strategy is not clearly explained and would benefit from stronger justification. The authors report testing only one- and three-factor models via CFA, yet EFA results are presented without any description of the procedure in the methods. In my opinion, a more rigorous approach would be to conduct EFA in Sample 1 to determine the optimal number of factors, followed by CFA in Sample 2 to confirm the structure. In addition, the CFA models tested show suboptimal fit indices (e.g., CFI < .900). I am therefore not convinced that the final model adequately fits the data. Relatedly, the criteria for model evaluation, construct validity testing, and measurement invariance thresholds should be more clearly justified and consistently applied.

Key results are underreported or presented unclearly. For example, chi-square statistics, degrees of freedom (rather than chi-square/df), and chi-square difference tests for invariance are not reported. RMSEA confidence intervals are missing despite being listed as evaluation criteria. Figures are difficult to interpret. Together, these issues make it difficult to evaluate the robustness of the findings.

Several psychometric concerns are also not adequately addressed. The Social Skills factor shows weak internal consistency (alpha = .619). Test-retest reliability values (r = .33-.62) are substantially lower than those reported in prior studies (Table 1) and warrant careful discussion. Partial scalar invariance across gender and language is acknowledged but not explained (e.g., why specific items function differently across subgroups). The reduction from 40 to 32 items may be acceptable given low factor loadings, but the implications for comparability with the original EQ should be discussed in more depth.

It is unclear how the total scores were calculated (e.g., in the Sex Differences in Empathy section). Did the authors use factor scores or simply sum the items? If unweighted scores were used, the Social Skills factor, with only four items, would have very little influence on the total score. The authors should clarify their approach and discuss the implications for interpreting both the overall M-EQ score and the three subscale scores. They should also address how the M-EQ scale is intended to be used: should it be treated as three separate constructs (cognitive, emotional, and social skills) or as a single construct (empathy)? Depending on how the authors define the measure, it may be necessary to examine a higher-order CFA model (one second-order factor representing empathy, with three first-order factors representing cognitive, emotional, and social skills).

Finally, the authors should provide the data used for analysis. If full data sharing is not possible, they should include sufficient information (e.g., item-level correlations, means, and standard deviations) in the supplemental materials.

6. PLOS authors have the option to publish the peer review history of their article (what does this mean?). If published, this will include your full peer review and any attached files.

Reviewer #1: **Yes:** Mahboubeh Dadfar, Ph.D., MPH

Reviewer #2: No

---

## [Author Response · Author response to Decision Letter 1]

12 Nov 2025

7 November 2025

Dear Editor and Reviewers,

Thank you for this feedback and to help improve on this manuscript. Please see our responses below. All page numbers and line numbers are based on the clean version. Thank you.

Best wishes,

Min

Reviewer #1:

Dear authors,

Thank you for your good manuscript, entitled "Assessing empathy in adults: A Malay language validation and measurement invariance of the Empathy Quotient (M-EQ)".

This is a valuable study. Here are my comments.

1. Title should change to “Empathy: Measurement invariance of the Empathy Quotient among Malaysian adults”.

*** We thank the reviewer for this suggestion. However, we would like to retain the existing title and not drop the ‘validation’ in the title because we had a 1-factor vs 3-factor structure analysed.

2. Keywords should change to “Empathy, Empathy Quotient (EQ), Measurement invariance, Psychometrics; Malaysia”.

*** We agree to drop two keywords; language type and gender. However we do not think it is appropriate to include ‘Malaysia’. The main interest is the Malay language and the Malay language is used in the Southeast Asia region, which meant that the M-EQ could be used in any of these countries.

3. Authors should add research questions/assumptions of their study in the end of Introduction section.

*** We thank the reviewer for this suggestion. We have now included the research question and hypotheses under Research Aim (see below and lines 91 to 99, and 118 to 119).

Research aim

At present, there is no Malay version of the EQ. The Malay language is the national language in Malaysia, Indonesia, Brunei, and Singapore with approximately 220 million speakers. It is also widely spoken in southern Thailand, Philippines, and East Timor [21]. For this study, our aim was to develop and validate a Malay language version of the EQ (M-EQ). Based on the inconsistent factor models reported in previous reviews (see Table 1 and [5]), our first objective was to examine whether the M-EQ aligns better with a 1-factor or a 3-factor structure model. Our second objective was to assess its temporal reliability, and thus we conducted a test-retest over a 4-week interval. Additionally, our third objective was to evaluate measurement invariance (MI) on two variables; language types and gender.

…affective items. Therefore, our fourth objective was to examine whether M-EQ measures empathy equally across both genders.

4. Authors should write all items for the EQ in the Table (All statements).

*** We seek further clarity about this. The M-EQ is located at the end of this manuscript if the reviewer is asking about this. We have included footnotes for the abbreviations in each table as well.

5. Authors should write clinical implications of their study.

*** We thank the reviewer for this suggestion. We have now added this into the manuscript. See line 398 to 413.

Although our study focused on neurotypical adult population, we wish to highlight the potential application of the M-EQ in clinical settings. Some proposed that the multifaceted concept of empathy could contribute to a more complete understanding of various clinical populations such as autism (Robinson, 2019). In their study, they used EQ alongside Autism Spectrum Quotient (AQ) and the Relatives Questionnaire (RQ) (retrospective, parent/informant-report version of the Childhood Autism Spectrum Test) to facilitate quicker identification of autistic individuals and reduce diagnostic wait times. Another study reported that people with low emotional empathy was associated with Dark Triad traits, while low cognitive empathy are specifically linked to narcissism and Machiavellianism (Turner et al., 2019), indictaing EQ’s potential utility for psychopathy assessments. Scoring low on cognitive empathy factor is also linked to poor social function in schizophrenia patients (Michaels et al., 2014). Note that in both studies (Michaels et al., 2014; Turner et al., 2019), the authors focused on specific factors (cognitive or emotional empathy) in the EQ, suggesting that clinicians could streamline assessments by selecting only the most relevant items for their target population without compromising diagnostic validity. These studies underscore the potential for the M-EQ to be adapted for the clinical population in Malaysia, offering a culturally relevant and time-efficient tool for psychological assessment.

6. Authors should write doi for all references.

*** We apologise for omitting the DOIs and have now included them in the Reference List.

Reviewer #2: I appreciate the authors’ effort to validate the Malay version of the Empathy Quotient (M-EQ) and to examine its measurement invariance across gender and language type. However, the manuscript is not yet easy to follow, and several conceptual, methodological, and reporting issues currently limit its scientific contribution and clarity.

1. The manuscript lacks conceptual coherence. The introduction emphasizes cognitive and affective empathy, but the final model includes three factors (cognitive, emotional, and social skills) without adequate justification for the additional factor. Terminology is also inconsistent (e.g., “emotional” vs. “affective” empathy; “sex” vs. “gender”), which reduces clarity. The rationale for including “social skills” as a separate factor and its theoretical grounding should be explained more thoroughly.

*** We thank the reviewer for this. We have now included further explanation on the 3-factor structure as well as more explanation on the “social skills” in both the Introduction and Discussion sections. We also reviewed the entire manuscript by removing ‘sex’ and replaced it with ‘gender’.

2. It is unclear how Table 1 was compiled. Did the authors conduct their own review, or was the table adapted from previous work? The methodology for obtaining the data in Table 1 should be clarified. In addition, the table would be more useful if it included information about the populations studied (e.g., clinical vs. general population), age ranges, and other relevant demographics.

*** We have added more details to Table 1 to include demographic details such as mean and SD for age in the whole sample. We checked the details and realised that we omitted two studies (Gouveia et al., 2012; Paolo Senese et al., 2018) and removed one study (Dimitrijević et al., 2012) because they tested adolescents as young as 15 years of age. We excluded Dimitrijevic et al. (2012) because empathy in adolescent is still developing rapidly due to individual emotional experience and regulatory processes as well as biological differences between boys and girls (Gaspar & Esteves, 2022).

3. The manuscript does not adequately justify the use of two samples. It is unclear why the authors treated the datasets as separate for some analyses and combined them for others (e.g., measurement invariance). In addition, because data collection was interrupted by COVID-19, the comparability of pre- and post-pandemic data should be examined. The authors should report whether participants recruited before versus after the pandemic differed on key demographic or psychometric variables, as such differences could affect the factor structure and reliability estimates.

*** We apologise for this confusion. Data was collected from two different samples at two distinct time points (2019 and 2022). We collected one round each for Sample 1 and Sample 2 participants. Note that Sample 2 participants completed the M-EQ twice after a four-week interval.

We had two separate data collection for methodological rigour. This is because Sample 1 (n = 232) was tested against the original 1-factor structure using confirmatory factor analysis (CFA). Our results showed that the one-factor model was a poor fit and therefore we performed exploratory factor analysis (EFA). Results showed that a 3-factor model was a better fit. To avoid capitalizing on chance and adhering to best practices, we did not confirm the new factor structure within the same dataset. Instead, we collected new data (Sample 2, n = 354) to conduct a CFA, which confirmed the 3-factor model. We would like to iterate that the factor structure was replicated consistently across both samples, which strengthens confidence in the robustness of the 3-factor solution despite the different collection periods. We have now made this cleared under Analytical Plan to describe our approach to the two data collection. Please refer to line 167 to 171.

We implemented a 3-step validation process. In Step 1, we tested a one-factor model using confirmatory factor analysis (CFA) with AMOS 27 in Sample 1 participants. We then conducted an exploratory factor analysis (EFA) using SPSS 26 to better understand the underlying structure (Step 2). Based on the EFA results, a three-factor model was proposed to which we tested this using CFA on Sample 2 participants to confirm its validity (Step 3).

We appreciate the comment regarding pre- and post-pandemic but this is not feasible due to our data collection phase. Nonetheless, we have addressed the influence of COVID-19 on our participants and have acknowledged this in the Limitations section. See line 390 to 397.

…we acknowledge that our data collection was impacted by the COVID-19 pandemic. Some researchers reported that although cognitive and emotional empathy remained intact during the pandemic, there were significant reduction in empathic social skills (Baiano et al., 2022; Ng et al., 2024), suggesting that interventions targeting social skills would be ideal in helping others cope with mental health challenges. Other studies reported that those who maintained physical distancing and wearing of masks were prosocial indicators which helps to minimise spreading the COVID-19 virus (Pfattheicher et al., 2020). Taken together, these studies suggest empathic response could possibly be atypical during the pandemic outbreak and may have continued long after as a new ‘norm’.

4. The analytical strategy is not clearly explained and would benefit from stronger justification. The authors report testing only one- and three-factor models via CFA, yet EFA results are presented without any description of the procedure in the methods. In my opinion, a more rigorous approach would be to conduct EFA in Sample 1 to determine the optimal number of factors, followed by CFA in Sample 2 to confirm the structure. In addition, the CFA models tested show suboptimal fit indices (e.g., CFI < .900). I am therefore not convinced that the final model adequately fits the data. Relatedly, the criteria for model evaluation, construct validity testing, and measurement invariance thresholds should be more clearly justified and consistently applied.

*** We agree with the suggestion raised by the reviewer and did conduct these analyses. We apologise for the lack of clarity. We have addressed this comment in the above question as well.

5. Key results are underreported or presented unclearly. For example, chi-square statistics, degrees of freedom (rather than chi-square/df), and chi-square difference tests for invariance are not reported. RMSEA confidence intervals are missing despite being listed as evaluation criteria. Figures are difficult to interpret. Together, these issues make it difficult to evaluate the robustness of the findings.

*** We apologise for omitting these details. Please see revised on line 247 to 265 and Table 4.

6. Several psychometric concerns are also not adequately addressed.

(a) The Social Skills factor shows weak internal consistency (alpha = .619).

*** We agree with the reviewer regarding the low internal reliability in the SS factor and have now expanded this in the Discussion. See line 296 to 312.

…We acknowledge that the Social Skills (SS) Cronbach alpha raises questionable internal consistency (0.619) but note that this factor in the M-EQ only has four items and it is known that scales with fewer items tend to have lower alpha values to reduced item covariance. In the SS factor, the largest factor loading was for item 27 “I get upset if I see people suffering on news programmes“ with a 0.740 while the lowest was item 24 “It upsets me to see animals in pain“ at 0.342. Item 29 “I can sense if I am intruding, even if the other person doesn’t tell me“ and item 40 “I can usually appreciate the other person’s viewpoint, even if I don’t agree with it“ had moderate loadings at 0.367 and at 0.419 respectively. Although the reliability is lower than ideal, similar findings were reported in other studies [9,11,16]. For example, the Turkish EQ version removed 11 items including items 27 and 29 because the factor loadings were below 0.30 and challenges in translating these items into local cultural context [6,12]. Others reported that EQ is positively correlated to social desirability scores because empathizing does require individuals to be compliant with expectations from others [10,12,16]. Taken together, researchers need to be aware of interpreting SS factor on its own with caution. Nonetheless, given the theoretical relevance of the SS factor and its contribution to the overall empathy construct, we retained this factor in the M-EQ.

(b) Test-retest reliability values (r = .33-.62) are substantially lower than those reported in prior studies (Table 1) and warrant careful discussion.

*** Regarding the test-retest reliability value, we apologise for this mistake. After checking our data, our M-EQ test-retest from Sample 2 is r = .794, p < .001. See line 278.

… correlation coefficients (0.794, p < 0.001). This indicated that the 32 items of the M-EQ have...

(c) Partial scalar invariance across gender and language is acknowledged but not explained (e.g., why specific items function differently across subgroups).

*** We have now expanded the MI results on the partial scalar invariance for both gender and language types. Please see revised on line 335 to 348.

… differently. For example, in the CE factor, Item 30 “People sometimes tell me that I have gone too far with teasing” suggests that males and females may have different baseline tendencies on how teasing is perceived. One study showed that boys initiated teasing significantly more often than girls – 78% versus 22% of the time [33], suggesting that teasing may be a social norm common among males. However, teasing can also be interpreted as a form of aggression. Studies have shown that girls display more relational aggression while boys display more physical and verbal aggression [34] but this differences tend to converge in adulthood, indicating that teasing works equally across both genders later on in life [35]. Similarly, the intercept invariance observed for Item 11 “I can pick up quickly if someone says one thing but means another” in EE could be explained by early gendered socialisation. Girls are typically more verbal with their friends while boys engage in more physical play and roughhousing [36]. This difference may give females an advantage in detecting subtle cues in conversations, potentially influencing how they respond to this item.

… types (Kaplan, 1966). It is also possible this item is not an indicator of empathy. The term “offended” (translated as kecil hati in Malay) carries varying emotional weight or social implications to the speaker depending on cultural context. Rather than reflecting empathic concern, responses may be shaped by individual personality traits or cultural norms around emotional expression. Additionally, the reference to a vague ‘someone’ is ambiguous and requires participants to subjectively assess the emotional significance of the remark.

(d) The reduction from 40 to 32 items may be acceptable given low factor loadings, but the implications for comparability with the original EQ should be discussed in more depth.

*** We have added this to Discussion. See lines 313 to 323.

… Further, the M-EQ should be used as per the original EQ [3] with similar Likert scales scoring to obtain a total empathy score. Compared to the original EQ instrument, the M-EQ has 32 items across a 3-factor model; the CE factor has 15 items, EE factor 13 items and SS four items. Lawrence et al. [4] had 11 items in the CE and EE respectively with 6 for SS and the Korean version also had similar 28 items [9], suggesting that the M-EQ i

---

## [Decision Letter · Decision Letter 1]

18 Dec 2025

PONE-D-25-12858R1Assessing empathy in adults: A Malay language validation and measurement invariance of the Empathy Quotient (M-EQ)PLOS One

Dear Dr. Yong,

Thank you for submitting your revised manuscript to PLOS ONE. After careful consideration, we feel that it has merit but does not fully meet PLOS ONE’s publication criteria as it currently stands. Therefore, we invite you to submit a revised version of the manuscript that addresses the points raised during the review process.

**The statistical analysis need to be revised and rechecked. Please see the reviewer' comments**

We look forward to receiving your revised manuscript.

Kind regards,

Juthatip Wiwattanapantuwong

Guest Editor

PLOS ONE

**Journal Requirements:**

Reviewers' comments:

Reviewer's Responses to Questions

**Comments to the Author**

1. If the authors have adequately addressed your comments raised in a previous round of review and you feel that this manuscript is now acceptable for publication, you may indicate that here to bypass the “Comments to the Author” section, enter your conflict of interest statement in the “Confidential to Editor” section, and submit your "Accept" recommendation.

Reviewer #2: (No Response)

2. Is the manuscript technically sound, and do the data support the conclusions?

Reviewer #2: Partly

3. Has the statistical analysis been performed appropriately and rigorously? 

Reviewer #2: No

4. Have the authors made all data underlying the findings in their manuscript fully available?

Reviewer #2: Yes

5. Is the manuscript presented in an intelligible fashion and written in standard English?

Reviewer #2: Yes

6. Review Comments to the Author

**Reviewer #2:** I appreciate the authors’ efforts in revising the manuscript. Overall, the authors have addressed many of my previous concerns (as Reviewer 2). However, based on the dataset and AMOS outputs provided on the OSF platform (https://osf.io/c872d/files/osfstorage), as well as my own examination of the data, I identified one major issue related to the statistical analyses that requires serious attention.

Specifically, the reported models appear to have been substantially modified, likely on modification indices. Such extensive post hoc model modification has long been recognized as risking capitalization on chance and may lead to incorrect or unstable models (MacCallum et al., 1992). I am therefore strongly concerned about this practice. Although post hoc model modification is not inherently inappropriate, each added path must be theoretically justified and clearly documented.

As shown in Table 4, the configural model has 828 degrees of freedom, whereas the baseline configural model without additional paths (e.g., correlated residuals and cross-loadings) should have 922 degrees of freedom. This discrepancy (922-828 = 94) indicates that 94 additional parameters (or paths) were introduced to improve model fit without being explicitly reported to readers, although they are visible in the AMOS files. I might be persuaded that such modifications are justified if modification indices from both datasets consistently indicated the need for the same additional paths. However, the AMOS outputs do not support this claim. Moreover, the models include implausible paths, such as relations between a latent factor and an item residual. Taken together, these issues lead me to conclude that the reported models are unacceptable.

Instead of adding numerous post hoc paths, I think it may be preferable to remove poorly performing items (provided this does not compromise validity). For example, the authors may select the best few items per factor and conduct a CFA without correlated residuals or cross-loadings, or include such parameters only when they are theoretically justified.

In addition, there are numerous statistical issues throughout the manuscript. I strongly recommend that the authors carefully re-examine their analytic approach and results, or consult an expert in factor analysis or structural equation modeling. The following issues are those I was able to identify:

- Lines 178-179 (tracked-changes version): “Modification index coefficients were used to examine cross-loadings between items.” Please clarify whether correlated residuals were also examined. In addition, cross-loadings reflect relationships between items and multiple factors, not relationships between items themselves.

- The current standard for factor retention is parallel analysis, rather than relying solely on eigenvalues greater than 1. Please report the results of a parallel analysis.

- Please re-evaluate whether principal axis factoring (PAF) is appropriate given that the data were reported as normally distributed, or whether maximum likelihood (ML) estimation would be more suitable.

- “Cattell’s scree plot” should be spelled with a double t (not “Catell”).

- Please provide a citation for the measurement invariance criteria (ΔCFI > -0.01, ΔRMSEA < 0.01, and ΔSRMR < 0.01).

- Please clarify the following statement: “Should configural invariance not be achieved, partial invariance was tested” Why was partial invariance considered only at the configural level?

- Lines 214-215: “The correlation values between the three factors were weak to moderate (-0.39 to 3.98).” Correlation coefficients should fall between -1 and 1.

- Lines 240-242: “χ²/df = 1.547, p < .001, CFI = 0.904, GFI = 0.895, AGFI = 0.846, TLI = 0.884, RMSEA = 0.039, SRMR = 0.055.” GFI and AGFI were not specified in the Analysis Plan section. Please also justify why model fit was deemed adequate when, for example, the TLI did not meet the stated cutoff.

- As noted in my previous review, please report the degrees of freedom for all tested models. Without df, it is not possible to evaluate the extent of model modification without access to the raw data.

- There should be multiple test-retest reliability values, one for the scale and one for each factor.

- Please clarify whether internal consistency estimates were examined separately for Sample 1 and Sample 2.

- Thank you for the clarification regarding the computation of total and subscale scores. However, given that the instrument is described as comprising three correlated first-order factors while also being used to generate a single total empathy score, it remains unclear whether the authors tested a second-order CFA model in which cognitive, emotional, and social skills factors load onto a single higher-order empathy construct. The unequal number of items across the three factors has important implications for the interpretation of the total score when unweighted sums are used. This issue requires explicit discussion in the manuscript.

In sum, substantial revisions are still required before this manuscript can be considered suitable for publication. I hope these comments will be helpful to the authors.

Reference: MacCallum, R. C., Roznowski, M., & Necowitz, L. B. (1992). Model modifications in covariance structure analysis: The problem of capitalization on chance. Psychological Bulletin, 111(3), 490–504. https://doi.org/10.1037/0033-2909.111.3.490

7. PLOS authors have the option to publish the peer review history of their article (what does this mean?). If published, this will include your full peer review and any attached files.

Reviewer #2: No

---

## [Author Response · Author response to Decision Letter 2]

19 Feb 2026

Dear Editor, Reviewer

Thank you for this feedback and it was much appreciated. Please see our responses below. All page numbers and line numbers are based on the clean version.

We have adopted the reviewer’s suggestions on changing our analytical approach and have now changed the almost the entire Results and Discussion sections to better reflect our findings.

Thank you.

Best wishes,

Min

***

Reviewer #2: I appreciate the authors’ efforts in revising the manuscript. Overall, the authors have addressed many of my previous concerns (as Reviewer 2). However, based on the dataset and AMOS outputs provided on the OSF platform (https://osf.io/c872d/files/osfstorage), as well as my own examination of the data, I identified one major issue related to the statistical analyses that requires serious attention.

Specifically, the reported models appear to have been substantially modified, likely on modification indices. Such extensive post hoc model modification has long been recognized as risking capitalization on chance and may lead to incorrect or unstable models (MacCallum et al., 1992). I am therefore strongly concerned about this practice. Although post hoc model modification is not inherently inappropriate, each added path must be theoretically justified and clearly documented.

As shown in Table 4, the configural model has 828 degrees of freedom, whereas the baseline configural model without additional paths (e.g., correlated residuals and cross-loadings) should have 922 degrees of freedom. This discrepancy (922-828 = 94) indicates that 94 additional parameters (or paths) were introduced to improve model fit without being explicitly reported to readers, although they are visible in the AMOS files. I might be persuaded that such modifications are justified if modification indices from both datasets consistently indicated the need for the same additional paths. However, the AMOS outputs do not support this claim. Moreover, the models include implausible paths, such as relations between a latent factor and an item residual. Taken together, these issues lead me to conclude that the reported models are unacceptable.

Instead of adding numerous post hoc paths, I think it may be preferable to remove poorly performing items (provided this does not compromise validity). For example, the authors may select the best few items per factor and conduct a CFA without correlated residuals or cross-loadings, or include such parameters only when they are theoretically justified.

In addition, there are numerous statistical issues throughout the manuscript. I strongly recommend that the authors carefully re-examine their analytic approach and results, or consult an expert in factor analysis or structural equation modeling. The following issues are those I was able to identify:

- Lines 178-179 (tracked-changes version): “Modification index coefficients were used to examine cross-loadings between items.” Please clarify whether correlated residuals were also examined. In addition, cross-loadings reflect relationships between items and multiple factors, not relationships between items themselves.

- The current standard for factor retention is parallel analysis, rather than relying solely on eigenvalues greater than 1. Please report the results of a parallel analysis.

- Please re-evaluate whether principal axis factoring (PAF) is appropriate given that the data were reported as normally distributed, or whether maximum likelihood (ML) estimation would be more suitable.

Response: Thank you for your feedback. We acknowledge that the statistical approach adopted in the previous version had limitations, particularly with factor retention and model specification. In response, we re-analysed the data following established best-practice recommendations for scale development and validation.

Please see page 13, line 161 to 192 under Analytical Approach.

Please see page 15, line 202 to page 20, line 265 under Factor Analysis. See below.

…We conducted EFA in sample 1 (n=232) in R. EFA was performed using maximum likelihood with oblimin rotation, given the expected correlations among factors. The data were suitable for factor analysis with KMO = 0.812, Bartlett p < .001. Factor retention was guided using Parallel Analysis. We used the following criteria: items were retained if they showed factor loading ≥ .40. Items were considered for removal when the items show cross-loading ≥ .30 on multiple factors and low communality (h² )< .20–.30 [30,31].

Parallel analysis initially supported a four-factor solution with an acceptable overall model quality (RMSEA= 0.037, RMSR = 0.05, TLI = 0.87). However, closer inspection of item‑level diagnostics indicated that this solution lacked psychometric and conceptual coherence. Several items have low communalities and some items have high cross-loading across factors indicating poor conceptual and psychometric coherence. Specifically, Factor 3 was defined by a single item (EQ27) with strong loading (λ = 0.82, h² = 0.68) while other items (EQ29, EQ40, and EQ39) showed weak loadings (0.32-0.36) and cross-loaded on Factor 4. A latent factor defined by only one item is considered psychometrically inadequate and was therefore deemed unacceptable. Similarly, items loading on factor 4 (EQ21, EQ22, EQ39 and EQ 25) showed weak factor loading (0.39 to 0.37 with cross loading), thus undermining this factor. Taken together, these findings suggest that the 4-factor solution represented an over extraction of factors and thus rejected.

We then removed seven items (EQ2, EQ3, EQ4, EQ17, EQ23, EQ24, and EQ25) for consistently low loadings across factors. EFA was re-estimated and parallel analysis was conducted again. Findings indicated a 3-factor solution with improved overall model quality (RMSEA= 0.041, RMSR= 0.05, TLI= 0.88). However, factor 3 remained psychometrically invalid with only EQ27 (factor loading of 1.00), indicating statistical over-extraction.

Given these findings, we evaluated a 2-factor solution and removed item EQ27 which was supported by parallel analysis. Model fit indices remained acceptable (RMSEA= 0.042, RMSR= 0.06, TLI= 0.87), and both factors were statistically acceptable. Factor 1 comprised of 13 items (EQ1, EQ11, EQ13, EQ14, EQ15, EQ22, EQ26, EQ28, EQ29, EQ34, EQ35, EQ36, and EQ38) with factor loadings ranging from 0.409-0.701. These 13 items were about the sharing an emotional state, a willingness to help others, and feeling distress or upset to another’s pain. An example was “I can pick up quickly if someone says one thing but means another – Item 11”. In sum, this factor was conceptualised as “emotional empathy” (EE). Factor 2 also comprised of 13 items (EQ5, EQ7, EQ8, EQ10, EQ12, EQ16, EQ18, EQ19, EQ20, EQ30, EQ31, EQ32, and EQ33) with factor loadings ranging from 0.411 to 0.580. These items contain perspective-taking, understand other’s people feelings, and imagined what was others’ perspectives. Example of an item is “I often find it difficult to judge if something is rude or polite – item 8”. Hence, this factor was conceptualised as “cognitive empathy” (CE).

Table 3. Factor loadings for M-EQ from EFA using the oblimin method (n = 232)

To confirm the structural validity of M-EQ with 26 items, we repeated CFA for 2-factor model with Sample 2 participants (n = 354) using weighted least squares mean‑ and variance‑adjusted (WLSMV) estimation. The model specification was informed by the EFA results, and 14 items previously identified as problematic were excluded prior to model estimation. Our results showed an adequate model of fit: χ²(298) = 671.96, p < .001; χ²/df = 2.25, CFI = 0.963, TLI = 0.960, RMSEA = 0.060 (90% CI .054–.066), SRMR = 0.074. Factor 1 (EE) was very strong, with all items loading above .50. Factor 2 (CE) showed acceptable fit, but three items (EQ10, 32, 33) exhibited very weak standardized loadings (< .30), indicating they did not reliably measure the latent construct in this sample. As such, these three items were removed and the CFA was re-estimated.

The refined two-factor model (23 items) demonstrated good fit to the data, χ²(229) = 528.40, p < .001; χ²/df = 2.31; CFI = .970; TLI = .967; RMSEA = .061 (90% CI [.054, .068]); SRMR = .072. Standardized factor loadings ranged from 0.51 to 0.77 for Factor 1 (EE) and from 0.35 to 0.76 for Factor 2 (CE). No post-hoc model modifications were applied. Although EQ19 showed a relatively lower standardized loading (0.353), it exceeded the minimum acceptable threshold and was retained to preserve content coverage of Factor 2. In addition, removing EQ19 did not result in a meaningful improvement in overall model fit.

We next estimated a second-order CFA model in which EE and CE factors were loaded onto a higher-order empathy construct using R (Lavaan package). The model did not converge to a proper solution, and produced a non‑invertible information matrix and a negative variance estimate for the second-order factor, indicating that the data does not support a unidimensional structure. Therefore, the correlated 2-factor model was retained for as the appropriate structural representation of the M-EQ, χ²(229) = 481.206, CFI = 0.973, TLI = 0.971, RMSEA = 0.056, SRMR = 0.070, with the latent factors showing a moderate negative correlation (r = −0.434, p < .001)….

***

“Cattell’s scree plot” should be spelled with a double t (not “Catell”).

Response: This is no longer applicable.

***

Please provide a citation for the measurement invariance criteria (ΔCFI > -0.01, ΔRMSEA < 0.01, and ΔSRMR < 0.01).

Response: This is no longer applicable as we have now changed the MI analytical approach to Item Response Theory-based Differential Item Functioning (DIF) analyses. Please see the revised section.

Please see page 20, line 267 to 288.

Measurement Invariance (MI) for M-EQ

Gender. Measurement invariance across gender was first examined using single‑group CFAs. Model fit was poor for males (CFI = 0.77) and inadequate for females (CFI = 0.86), indicating that configural invariance could not be established and thus multi-group CFA invariance testing was not conducted. Instead, gender-related measurement equivalence was examined using Item Response Theory-based Differential Item Functioning (DIF) analyses. First, we examined a posteriori (EAP) reliability for both subscales and it was high for EE (0.865) and acceptable for CE (0.775), supporting the use of latent scores for group comparisons. DIF analyses revealed no evidence of gender-related DIF for either EE or CE. Likelihood-ratio DIF tests were conducted for discrimination (a1) and threshold (d1–d3) parameters, with p-values adjusted using the Benjamini–Hochberg false discovery rate procedure. No items showed statistically significant DIF and none of the drop‑in‑fit tests indicated meaningful improvements in model fit. These indicate invariant item functioning across gender groups (see Table 4).

Language types. We repeated the analyses using the sample 2 (Latin: English/Malay = 234, non-Latin: Chinese/Tamil = 120). EAP reliability estimates again indicated adequate measurement precision for latent mean comparisons (EE = 0.862; CE = 0.767). DIF analyses showed no evidence of differential item functioning for either EE or CE across language groups. Likelihood-ratio DIF tests for discrimination (a1) and threshold (d1–d3) parameters, with Benjamini–Hochberg FDR correction, yielded no significant item-level differences (all Δ−2LL = 0). These results indicate invariant item functioning across language types, suggesting that M -EQ items function equivalently for respondents using Latin and non-Latin as their first language (see Table 4).

***

Please clarify the following statement: “Should configural invariance not be achieved, partial invariance was tested” Why was partial invariance considered only at the configural level?

Response: This is no longer applicable as we have now changed the MI analytical approach. See above.

***

- Lines 214-215: “The correlation values between the three factors were weak to moderate (-0.39 to 3.98).” Correlation coefficients should fall between -1 and 1.

Response: We acknowledge this mistake. However, this is no longer applicable because we have now changed the approach as recommended by the reviewer.

***

- Lines 240-242: “χ²/df = 1.547, p < .001, CFI = 0.904, GFI = 0.895, AGFI = 0.846, TLI = 0.884, RMSEA = 0.039, SRMR = 0.055.” GFI and AGFI were not specified in the Analysis Plan section. Please also justify why model fit was deemed adequate when, for example, the TLI did not meet the stated cutoff. –

Response: This is no longer applicable because we have now changed the approach as recommended by the reviewer.

***

As noted in my previous review, please report the degrees of freedom for all tested models. Without df, it is not possible to evaluate the extent of model modification without access to the raw data.

Response: We apologize for omitting the degrees of freedom. We have now included this detail across the Results section.

** see page 19, line 196 - 198.

… Results showed an inadequate model of fit for 1-factor: χ² (740) = 1901.8, p < .001, χ²/df = 2.57, RMSEA = 0.082, SRMR = 0.1177; CFI = .449; TLI = .419. These …

** see page 18, line 245 - 246.

… results showed an adequate model of fit: χ²(298) = 671.96, p < .001; χ²/df = 2.25, CFI = 0.963, TLI = 0.960, RMSEA = 0.060 (90% CI .054–.066), SRMR = 0.074. Factor 1 (EE) is very strong…

** see page 18, line 259 - 253.

The refined two-factor model (23 items) demonstrated good fit to the data, χ²(229) = 528.40, p < .001; χ²/df = 2.31; CFI = .970; TLI = .967; RMSEA = .061 (90% CI [.054, .068]); SRMR = .072.

***

There should be multiple test-retest reliability values, one for the scale and one for each factor.

Response: We agree with the reviewer. We have now reported both reliability values for Time (Time 1 and Time 2), and scores (per subscale: EE, CE, and total M-EQ).

** see page 23, line 307 - 311.

… Test–retest reliability was examined using Pearson correlations between Time 1 and Time 2 scores. The EE subscale demonstrated good temporal stability (r = 0.734, p < .001), as did the CE subscale (r = 0.748, p < .001). The total empathy score showed moderate test–retest reliability (r = 0.805, p < .001). Overall, these findings indicate satisfactory temporal stability, particularly for the subscale scores.

***

Please clarify whether internal consistency estimates were examined separately for Sample 1 and Sample 2.

Response: We performed internal consistency for Sample 2 only using Cronbach’s alpha (α) and McDonald’s omega (ωₜ).

** see page 23, line 298 – 306.

Internal consistency of the M-EQ 23 items was evaluated using both Cronbach’s α and McDonald’s ωₜ. The EE subscale demonstrated good reliability (α = .86, ωₜ = .86), and the CE subscale showed acceptable reliability (α = .78, ωₜ = .78). As the scale exhibited a multidimensional structure, Cronbach’s alpha for the total score was lower (α = .67), which is expected for scales comprising multiple correlated dimensions, as alpha assumes unidimensionality and equivalent factor loadings. When these assumptions are violated, alpha tends to underestimate reliability [32,33]. Therefore, McDonald’s omega was also computed for the total score, yielding high composite reliability (ωₜ = .88). Subscale scores are emphasized for interpretation, with the total score reported as a supplementary index.

***

Thank you for the clarification regarding the computation of total and subscale scores. However, given that the instrument is described as comprising three correlated first-order factors while also being used to generate a single total empathy score, it remains unclear whether the authors tested a second-order CFA model in which cognitive, emotional, and social skills factors load onto a single higher-order empathy construct. The unequal number of items across the three factors has important implications for the interpretation of the total score when unweighted sums are used. This issue requires explicit discussion in the manuscript. In sum, substantial revisions are still required before this manuscript can be considered suitable for publication. I hope these comments will be helpful to the authors.

Response: We thank the reviewer for raising the importan

---

## [Decision Letter · Decision Letter 2]

10 Mar 2026

PONE-D-25-12858R2Assessing empathy in adults: A Malay language validation and measurement invariance of the Empathy Quotient (M-EQ)PLOS One

Dear Dr. Yong,

Thank you for submitting your manuscript to PLOS ONE. After careful consideration, we feel that it has merit but does not fully meet PLOS ONE’s publication criteria as it currently stands. Therefore, we invite you to submit a revised version of the manuscript that addresses the points raised during the review process.

I do agree with reviewer 2 that more details of explanation to change the analysis from Multigroup CFA to IRT based MI are required and also some words about the removed items should be additionally discussed.

We look forward to receiving your revised manuscript.

Kind regards,

Juthatip Wiwattanapantuwong

Guest Editor

PLOS One

Journal Requirements:

Additional Editor Comments:

- Please provide more details and substantial explanations and references of why Item Response Theory-based Differential Item Functioning (DIF) analyses were used instead of multi-group SEM (P. 29 and 34).

- Please discuss about the omitted items from the scale to demonstrate that no essential elements have been removed.

- The first objective "our first objective was to examine whether the M-EQ aligns better with a 1-factor or a 3-95 factor structure model." should be revised to fit with the current finding.

Reviewers' comments:

Reviewer's Responses to Questions

**Comments to the Author**

1. If the authors have adequately addressed your comments raised in a previous round of review and you feel that this manuscript is now acceptable for publication, you may indicate that here to bypass the “Comments to the Author” section, enter your conflict of interest statement in the “Confidential to Editor” section, and submit your "Accept" recommendation.

Reviewer #2: (No Response)

2. Is the manuscript technically sound, and do the data support the conclusions?

Reviewer #2: Partly

3. Has the statistical analysis been performed appropriately and rigorously? 

Reviewer #2: No

4. Have the authors made all data underlying the findings in their manuscript fully available?

Reviewer #2: Yes

5. Is the manuscript presented in an intelligible fashion and written in standard English?

Reviewer #2: Yes

6. Review Comments to the Author

Reviewer #2: I appreciate the substantial effort the authors have made in revising the manuscript. The paper is clearly improved, and the overall model-building strategy is now much clearer than in the previous version.

That said, my main concern for this revision is the new measurement invariance analysis. The results showed that the configural multigroup CFA did not fit well, thus measurement invariance was not established. From my understranding, however, the authors then shifted to IRT-based DIF analyses for the two subscales (EE, CE) separately, which changes the model being tested rather than resolving the original issue that the 2-factor model did not fit the data well across groups. I do not see a clear theoretical justification for analyzing the subscales separately. Therefore, I do not think the current evidence is sufficient to support a broad claim of measurement invariance.

Because the files on OSF do not allow full replication of the reported results, I can only infer what may have occurred. Typically, if a single-group CFA model fits the data excellently, the corresponding multigroup model should also show at least acceptable fit, which does not appear to be the case here. One possibility is that the single-group analyses used WLSMV, which may yield more favorable fit indices in some situations (Xia & Yang, 2019), whereas the multigroup analyses may have relied on maximum likelihood, resulting in substantially poorer fit. It is also important to note that the procedure for testing measurement invariance with WLSMV differs from that used in standard CFA with maximum likelihood (Sass, Schmitt, & Marsh, 2014). Thus, if the authors would like to return to the CFA framework for measurement invariance using the WLSMV estimator, they should follow established best practices. If the authors instead wish to use IRT, they should do so consistently and describe the technique more thoroughly so that readers can clearly understand the analytic approach.

Moreover, I encourage the authors to strengthen the limitations section. First, they should explicitly acknowledge the extent of model refinement that occurred during the revision process, including the iterative and partly data-driven nature of the final 2-factor solution. Second, the manuscript currently provides limited evidence of validity beyond factor analyses. The revised paper relies mainly on EFA, CFA, and IRT-based analyses, and even these provide only partial support for the revised measurement model. This should be stated more clearly as a limitation. In addition, the overall language of the manuscript should be softened so that the conclusions more accurately reflect the strength of the evidence. Relatedly, the authors should carefully review the abstract, methods, results, discussion, and tables for consistency, as the analytic strategy changed substantially and all sections should align with the revised model and its limitations. It may also be helpful to include figures illustrating the mesurement models.

In summary, although the manuscript is substantially improved, important concerns remain, particularly regarding the measurement invariance analysis. Addressing these issues may substantially change the study’s conclusions.

7. PLOS authors have the option to publish the peer review history of their article (what does this mean?). If published, this will include your full peer review and any attached files.

Reviewer #2: No

---

## [Author Response · Author response to Decision Letter 3]

17 Apr 2026

16 April, 2026

Dear Editor and Reviewer,

Thank you for this feedback and it was much appreciated. Please see our responses below. All page numbers and line numbers are based on the clean version.

We have adopted the reviewer’s suggestions on changing our analytical approach and have now changed the almost the entire Results and Discussion sections to better reflect our findings.

Thank you.

Best wishes,

Min

***

Additional Editor Comments:

- Please provide more details and substantial explanations and references of why Item Response Theory-based Differential Item Functioning (DIF) analyses were used instead of multi-group SEM (P. 29 and 34).

Response: Thank you for your feedback. This was raised by Reviewer 2 as well and our response is included under the Reviewer 2 below.

- Please discuss about the omitted items from the scale to demonstrate that no essential elements have been removed.

Response: From the original 40 items, we now have 23 items which meant that we omitted 17 items in total. While this number appears high, Muncer and Ling (2006) had 15 items in their revised version and this was supported by several others (Gouveia et al., 2012; Paolo Senese et al., 2018; Zhao et al., 2018).

As per our Discussion (see p. 24 - 26), the items identified as cognitive empathy, emotional empathy and social skills domains were different across languages. This suggests that interpretation of these items could be a result of the language. The variability across language versions suggests that certain items may be more sensitive to linguistic and cultural interpretation rather than core empathic capacity per se, supporting their removal in the interest of cross-cultural validity.

We believe that our robust analytical approach with the EFA, CFA implemented is sound psychometrically. The omitted items predominantly consisted of those with weak factor loadings, substantial cross-loadings across domains, or low communalities, indicating limited contribution to the latent constructs in the Malay language context. Further, the omitted items did not represent unique conceptual facets of empathy that were absent from the retained items; rather, they overlapped substantially in content with retained indicators within each domain. One clear difference between M-EQ and other studies is the absence of the Social Skills (SS) domain. The SS domain has often been reported with low internal consistency with some indicating that this domain is related to social desirability (Berthoz et al., 2008; Kim & Lee, 2010; Kose et al., 2018; Preti et al., 2011).

In addition to statistical criteria, the final item set was evaluated through expert review by two bilingual authors with subject-matter expertise, who independently confirmed that the retained items adequately capture the essential conceptual content of each empathy domain.

***

… other languages reported a 1-factor (Zhao et al., 2018), 4-factor (Zhang et al., 2018) but the majority showed a 3-factor model with EE, CE, and SS (Berthoz et al., 2008; Gouveia et al., 2012; Groen et al., 2015; Kim & Lee, 2010; Kose et al., 2018; Kosonogov, 2014; Paolo Senese et al., 2018; Preti et al., 2011). Yet some acknowledged issues pertaining to SS and EE. One study showed multiple loading on the same item across two factors (Kose et al., 2018) and another suspected that the SS factor is better suited to measure personality type i.e. assertiveness rather than an empathy dimension (Gouveia et al., 2012). Others reported lower consistency and…

… items in their analysis but others identified different items for CE, EE and SS respectively (Groen et al., 2015; Preti et al., 2011), indicating interpretation of these items may mean differently due to subtle nuances in the translated language, rather than core empathic capacity per se. Taken together, we are confident that our analyses for the 23-item M-EQ is robust psychometrically and very much aligned with theoretical underpinning empathy dimensions. Importantly, all core domains and subcomponents of the original EQ framework remain represented in the final item set, with each domain retaining multiple indicators reflecting its theoretical breadth. The omitted items predominantly consisted of those with weak factor loadings, substantial cross-loadings across domains, or low communalities, indicating limited contribution to the latent constructs in the Malay language context. Further, the omitted items did not represent unique conceptual facets of empathy that were absent from the retained items; rather, they overlapped substantially in content with retained indicators within each domain. As such, item removal reflects refinement of measurement rather than loss of conceptual content, ensuring preservation of essential empathic elements while improving clarity and cultural suitability….

- The first objective "our first objective was to examine whether the M-EQ aligns better with a 1-factor or a 3-factor structure model." should be revised to fit with the current finding

Response: We agree that we need to change this sentence, but we disagree in revising to fit the current finding. We have now changed this in Abstract, Introduction and Discussion to better reflect our position.

Reviewer #2:

I appreciate the substantial effort the authors have made in revising the manuscript. The paper is clearly improved, and the overall model-building strategy is now much clearer than in the previous version.

That said, my main concern for this revision is the new measurement invariance analysis. The results showed that the configural multigroup CFA did not fit well, thus measurement invariance was not established. From my understanding, however, the authors then shifted to IRT-based DIF analyses for the two subscales (EE, CE) separately, which changes the model being tested rather than resolving the original issue that the 2-factor model did not fit the data well across groups. I do not see a clear theoretical justification for analyzing the subscales separately. Therefore, I do not think the current evidence is sufficient to support a broad claim of measurement invariance.

Because the files on OSF do not allow full replication of the reported results, I can only infer what may have occurred. Typically, if a single-group CFA model fits the data excellently, the corresponding multigroup model should also show at least acceptable fit, which does not appear to be the case here. One possibility is that the single-group analyses used WLSMV, which may yield more favorable fit indices in some situations (Xia & Yang, 2019), whereas the multigroup analyses may have relied on maximum likelihood, resulting in substantially poorer fit. It is also important to note that the procedure for testing measurement invariance with WLSMV differs from that used in standard CFA with maximum likelihood (Sass, Schmitt, & Marsh, 2014). Thus, if the authors would like to return to the CFA framework for measurement invariance using the WLSMV estimator, they should follow established best practices. If the authors instead wish to use IRT, they should do so consistently and describe the technique more thoroughly so that readers can clearly understand the analytic approach.

Response: We thank the reviewer for the constructive feedback regarding the measurement invariance analyses. We agree that clarity and methodological consistency are essential in evaluating cross-group equivalence, and we have substantially revised this section of the manuscript accordingly.

First, we re-examined all measurement invariance analyses using a consistent CFA framework in R (lavaan), applying an estimator appropriate for ordinal data using WLSMV (see Methods – Analytical Plan p.15). In the previous version, the previous version included a mixture of WLSMV (single‑group) and ML-based estimation (multigroup), which may have artificially inflated differences in model fit. These inconsistencies have now been fully resolved. In summary, the revised analyses show acceptable single-group fit for gender, acceptable configural model fit across groups, and that there is support for metric invariance based on recommended ΔCFI and ΔRMSEA thresholds although there is some elevation in SRMR, which we then discussed this in the Discussion.

Second, we clarify that we it was not our intention to shift from CFA to IRT as an alternative modelling approach. In the revised manuscript, multigroup CFA remains the primary method to evaluate measurement invariance. The results now show that the configural model demonstrates acceptable overall fit, and metric invariance is supported based on established criteria (ΔCFI and ΔRMSEA) with some degree of model misfit remains (e.g., elevated SRMR values) (see Results on p.20 to 21). We also now provide a cautious interpretation, avoiding any broad claim of full invariance given the slightly elevated SRMR values in some groups.

To avoid confusion, we now clearly state that the IRT-based DIF analyses were conducted as supplementary item-level diagnostics rather than as a replacement for CFA-based invariance testing. The purpose was to examine whether the modest misfit in some groups (e.g. in non-Latin language group) could be attributed to specific items functioning differently. Importantly, the DIF analyses were conducted within each subscale (EE and CE) to align with the established two-factor structure, rather than to redefine the measurement model.

***

… Once the construct validity of the M-EQ is determined, we evaluated measurement invariance following a 2-stage analytic strategy grounded primarily in CFA. We conducted multi-group CFA to examine invariance across gender and language types using WLSMV estimator in R (lavaan package). Measurement invariance was assessed sequentially. First, we tested configural invariance to determine whether the two‑factor structure was comparable across groups. Next, we tested metric invariance by constraining factor loadings to equality. Model fit was evaluated using CFI, RMSEA, and SRMR, and changes in fit were judged using recommended criteria (ΔCFI ≤ .010, ΔRMSEA ≤ .015, ΔSRMR ≤ 0.030) (Chen, 2007). Given known sensitivity of SRMR in WLSMV models, this index was interpreted cautiously.

To supplement the CFA results and to explore potential item-level sources of misfit, we conducted item response theory–based differential item functioning (DIF) analyses. These analyses were intended as diagnostic tools to identify whether specific items functioned differently across groups. Separate multigroup graded response models were estimated for the EE and CE subscales to align with the established two‑factor structure. Anchor items were selected a priori and used to link groups to a common latent metric. DIF was evaluated using likelihood‑ratio tests comparing nested models, with Benjamini–Hochberg correction applied to control for false discovery. Expected a posteriori (EAP) reliability was computed to assess the precision of latent trait estimates.

Gender. Measurement invariance across gender was first examined using single‑group CFAs. Single-group CFA indicated acceptable fit in the female group (CFI = 0.975, TLI = 0.972, RMSEA = 0.061, SRMR = 0.080), and the male group also showed acceptable CFI, TLI, and RMSEA values but elevated SRMR (CFI = 0.944, TLI = 0.938, RMSEA = 0.067, SRMR = 0.103). We next proceed with the configural model, and it demonstrated acceptable overall fit across gender (CFI = 0.968, TLI = 0.965, RMSEA = 0.063, SRMR = 0.088), indicating that the two-factor structure was broadly similar across male and female participants. Constraining factor loadings to equality across gender resulted in acceptable model fit (CFI = 0.959, RMSEA = 0.070, SRMR = 0.093), with minimal changes in fit indices (ΔCFI = −0.009, ΔRMSEA = +0.007), supporting metric invariance. However, the slightly elevated SRMR values warrant cautious interpretation. We further examined gender-related item level equivalence using Item Response Theory-based Differential Item Functioning (DIF) analyses. First, we examined a posteriori…

Language types. We repeated the analyses using the sample 2 (Latin: English/Malay = 234, non-Latin: Chinese/Tamil = 120). Single-group CFA indicated acceptable fit for the Latin group (CFI = 0.964, RMSEA = 0.062, SRMR= 0.082), whereas the non-Latin group showed acceptable but slightly poorer fit for CFI particularly in terms of RMSEA and SRMR (CFI = 0.957, RMSEA = 0.089, SRMR = 0.108). The configural model demonstrated acceptable overall fit (CFI = 0.961, RMSEA = 0.072, SRMR = 0.091), indicating that the two-factor structure was broadly similar across language groups, although model fit was not optimal. Constraining factor loadings to equality resulted in minimal changes in model fit (metric model: CFI = 0.953, RMSEA = 0.077, SRMR = 0.096; ΔCFI = −0.008, ΔRMSEA = +0.005), suggesting support for metric invariance across language groups. However, given the elevated SRMR values and poorer fit in the non-Latin group, these findings should be interpreted with caution. DIF analyses were conducted to further examine potential item-level non-invariance. EAP reliability estimates again indicated adequate measurement precision for latent mean comparisons (EE = 0.862; CE = 0.767). DIF analyses showed no evidence of differential item…

---

Moreover, I encourage the authors to strengthen the limitations section. First, they should explicitly acknowledge the extent of model refinement that occurred during the revision process, including the iterative and partly data-driven nature of the final 2-factor solution. Second, the manuscript currently provides limited evidence of validity beyond factor analyses. The revised paper relies mainly on EFA, CFA, and IRT-based analyses, and even these provide only partial support for the revised measurement model. This should be stated more clearly as a limitation. In addition, the overall language of the manuscript should be softened so that the conclusions more accurately reflect the strength of the evidence. Relatedly, the authors should carefully review the abstract, methods, results, discussion, and tables for consistency, as the analytic strategy changed substantially and all sections should align with the revised model and its limitations. It may also be helpful to include figures illustrating the mesurement models.

Response: We agree with the reviewer. We have now revised Limitations section (see p.29 to 30).

… We acknowledge that there are several limitations in our study. First, the final two‑factor, 23‑item structure of the M‑EQ emerged through an iterative process that included both theory‑driven reasoning and data‑driven refinement. Although this approach is common in early-stage instrument validation, it raises the possibility that the model may partly reflect sample‑specific characteristics. Although we have a large sample, we acknowledge that our sampling was not sufficiently diverse to represent the human lifespan because our sample is entirely young adults. Further, our university student sampling limits the generalisability to the wider population. Another limitation is that we did not directly measure intelligence. While the EQ does require some verbal intelligence, 99.4% of our sample completed a minimum of 12 years of formal education, suggesting that they do have at least average intelligence.

Second, although the study provides initial evidence for construct validity based on EFA, CFA, and supplementary IRT‑based DIF analyses, the validity evidence is limited in scope because our evaluation focused primarily on the internal structure. We did not include other behavioural measures such as emotion recognition tasks (Besel & Yuille, 2010) or other empathy scales such as Toronto Empathy Questionnaire (Spreng et al., 2009) to assess convergent, discriminant or predictive validity. As such, conclusions regarding the construct validity of the M‑EQ should be considered preliminary and strengthened in future work using multi‑method validation approaches and across different samples. While the measurement invariance results support broad comparability of the M‑EQ across gender and language groups; the evidence is not unequivocal. While configural and metric invariance were supported based on recommended fit index

---

## [Editor Report · Decision Letter 3]

29 Apr 2026

Assessing empathy in adults: A Malay language validation and measurement invariance of the Empathy Quotient (M-EQ)

PONE-D-25-12858R3

Dear Dr. Yong,

We’re pleased to inform you that your manuscript has been judged scientifically suitable for publication and will be formally accepted for publication once it meets all outstanding technical requirements.

Kind regards,

Juthatip Wiwattanapantuwong

Guest Editor

PLOS One
---

## [Editor Report · Acceptance letter]

PONE-D-25-12858R3

PLOS One

Dear Dr. Yong,

I'm pleased to inform you that your manuscript has been deemed suitable for publication in PLOS One. Congratulations! Your manuscript is now being handed over to our production team.

Kind regards,

on behalf of

Dr. Juthatip Wiwattanapantuwong

Guest Editor

PLOS One